# Harnessing the acceptor substrate promiscuity of *Clostridium botulinum* Maf glycosyltransferase to glyco-engineer mini-flagellin protein chimeras
Sonali Sunsunwal, Aasawari Khairnar, Srikrishna Subramanian & T.N.C. Ramya ✉

Several bacterial flagellins are O-glycosylated with nonulosonic acids on surface-exposed Serine/Threonine residues by Maf glycosyltransferases. The *Clostridium botulinum* Maf glycosyltransferase (*Cb*Maf) displays considerable donor substrate promiscuity, enabling flagellin O-glycosylation with N-acetyl neuraminic acid (Neu5Ac) and 3-deoxy-D-manno-octulosonic acid in the absence of the native nonulosonic acid, a legionaminic acid derivative. Here, we have explored the sequence/structure attributes of the acceptor substrate, flagellin, required by *Cb*Maf glycosyltransferase for glycosylation with Neu5Ac and KDO, by co-expressing *C. botulinum* flagellin constructs with *Cb*Maf glycosyltransferase in an *E. coli* strain producing cytidine-5'-monophosphate (CMP)-activated Neu5Ac, and employing intact mass spectrometry analysis and sialic acid-specific flagellin biotinylation as readouts. We found that *Cb*Maf was able to glycosylate mini-flagellin constructs containing shortened alpha-helical secondary structural scaffolds and reduced surface-accessible loop regions, but not non-cognate flagellin. Our experiments indicated that *Cb*Maf glycosyltransferase recognizes individual Ser/Thr residues in their local surface-accessible conformations, in turn, supported in place by the secondary structural scaffold. Further, *Cb*Maf glycosyltransferase also robustly glycosylated chimeric proteins constructed by grafting cognate mini-flagellin sequences onto an unrelated beta-sandwich protein. Our recombinant engineering experiments highlight the potential of *Cb*Maf glycosyltransferase in future glycoengineering applications, especially for the neo-O-sialylation of proteins, employing *E. coli* strains expressing CMP-Neu5Ac (and not CMP-KDO).

Flagellin, the monomeric unit of the bacterial motility appendage, the flagellum, is a Γ (capital Greek letter Gamma) - shaped protein with the vertical rod-shaped segment comprising the terminal alpha-helical domains, D0 and D1, and the horizontal segment comprising the central hypervariable surface-exposed domains, D2 and D3. Ser/Thr O-glycosylation has been observed in this surface-exposed D2/D3 domain region or in the D1 loop region (in flagellins lacking discrete D2/D3 domains) in several Gram-negative and Gram-positive bacteria[1,2] but no conserved sequon for flagellin O-glycosylation is known. The non-sialic acid nonulosonic acids (9-carbon sugars with a carboxyl group at C1 and an exocyclic chain from C7 to C9[3]), pseudaminic acid and legionaminic acid and their derivatives, are the common glycan moieties[4–18]. Flagellin modification with these glycans is catalyzed by the Maf (Motility Accessory Factor)[7,19–23] and the recently identified Flm flagellin glycosyltransferases[24,25]. In bacteria that do possess these flagellin glycosyltransferases, *maf* or *flm* gene inactivation results in loss of flagellin glycosylation, flagellin export, flagellar assembly, and bacterial cell motility[7,19–25]. The single, publically available X-ray diffraction solved crystal structure of Maf glycosyltransferase from *Magnetospirillum magneticum* comprises three domains, of which the central Maf_flag10 domain (pfam01973) shares structural similarity with Cst-II (a sialyl-transferase of *C. jejuni*)[23]. The N-terminal domain adopts a degenerated Rossmann-like fold with five alpha helices flanking parallel beta-sheets of four beta strands, and the C-terminal domain is composed mainly of alpha-helical bundles. The function of these domains remains unclear, although the N-terminal domain shows weak similarities with some motifs in methyltransferases and the C-terminal domain shows weak similarities with

CSIR- Institute of Microbial Technology, Sector 39-A, Chandigarh, 160036, India. ✉e-mail: ramya@imtech.res.in

flagellin export chaperones (such as FliS in *Aquifex aeolicus*) and with flagellins (such as FliC in *Burkholderia pseudomallei*)[23]

Maf glycosyltransferases display donor substrate promiscuity. *Campylobacter jejuni* Maf glycosyltransferases specific for pseudaminic acid and legionaminic acid accept CMP-azido- pseudaminic acid and CMP-azido-legionaminic acid, respectively, as donor substrates[26,27]. Further, the *Clostridium botulinum* Maf legionaminic acid glycosyltransferase and the *Geobacillus kaustophilus* Maf glycosyltransferase (predicted to be a legionaminic acid glycosyltransferase by genetic profiling experiments[25]) accept cytidine-5'-monophosphate (CMP)-activated N-acetyl neuraminic acid (Neu5Ac) and 3-deoxy-D-manno-octulosonic acid or keto-deoxyoctulosonate (KDO) as donor substrates[18], in the absence of the native donor. However, the acceptor substrate promiscuity of Maf glycosyltransferases has not been studied albeit a few reports indicate the molecular features of the acceptor substrate, flagellin, that are important for recognition by Maf glycosyltransferases. One, Maf glycosyltransferases modify Ser/Thr residues in the surface-exposed D2/D3 or D1 loop domain region; two, the *Aeromonas caviae* Maf glycosyltransferase modifies Ser/Thr residues flanked by apolar amino acids like leucine and isoleucine[28]; and three, the *A. caviae* Maf glycosyltransferase interacts with both unglycosylated and glycosylated flagellin[29].

Here, we used *C. botulinum* flagellin (*Cb*Fla) deletion constructs in order to identify the molecular features of flagellin critical for recognition and/or modification by the cognate Maf glycosyltransferase (*Cb*Maf). We also explored the acceptor substrate promiscuity of Maf glycosyltransferases by presenting non-cognate flagellin proteins and mini-flagellin protein chimeras as acceptor substrates. Our experiments indicate that CbMaf glycosyltransferase can glycosylate mini-flagellin constructs and chimeras, but not non-cognate flagellin proteins, with Neu5Ac and KDO, likely by recognizing individual Ser/Thr residues in their local surface-accessible conformations that are supported in place by the secondary structural scaffold. Our findings indicate that CbMaf glycosyltransferase might be useful for the neo-O-sialylation of recombinant proteins, such as biotherapeutics, in *E. coli* strains that overexpress CMP-Neu5Ac and not CMP-KDO.

## Results

### Model system for identifying the minimal sequence/structural features of *Cb*Fla required for recognition and/or modification by *Cb*Maf

Our model system involved heterologous expression of the recombinant *C. botulinum* F Str. Langeland flagellin constructs with or without the cognate (*C. botulinum*) Maf[18] in the *E. coli* EV136 strain[30,31], a *neuS* defective *E. coli* strain, which accumulates intracellular CMP-Neu5Ac that acts as donor substrate for *Cb*Maf (Fig. 1a).

The full-length *Cb*Fla protein comprises the D0 and D1 domains. The D0 domain is composed of the N-terminal alpha-helical D0 domain (ND0) and the C-terminal alpha-helical D0 domain (CD0). The D1 domain is composed of two N-terminal helices (ND1a and ND1b), a C-terminal helix (CD1), and a loop region (corresponding to the reduced D2/D3 region, and referred to by us as the D1 loop region) that bridges the ND1b and the CD1 helices of the D1 domain (Fig. 1b, c). In order to identify the minimal sequence/structural context of *Cb*Fla required for recognition and/or modification by *Cb*Maf, we designed *Cb*Fla constructs that contained varying lengths of the D1 loop region and/or the D0/D1 helices (construct and oligonucleotide primer sequences provided in Supplementary Data 1, Supplementary Tables S1, 2). We expressed these flagellin constructs singly or together with *Cb*Maf in *E. coli* EV136, and assessed their glycosylation status by employing a sialic acid-specific labeling procedure (PAL; periodate oxidation followed by ligation with aminooxy-biotin[32]) or intact mass spectrometry analysis.

We have previously demonstrated glycosylation of full-length *Cb*Fla on Ser/Thr residues in and around the D1 loop region by co-expressing *Cb*Maf in *E. coli* EV136 cells cultured in LB medium. Considering the different expression vectors (S-tagged *Cb*Maf expressed from pCDF-Duet-1 vector)

and culture medium (M9 minimal medium) used in this study for expressing the *Cb*Fla constructs, we first assessed glycosylation of full-length hexahistidine-tagged *Cb*Fla, expressed singly or together with *Cb*Maf in *E. coli* EV136 cells cultured in M9 minimal medium and purified by Ni-NTA metal ion chromatography (Fig. 1d–f and Supplementary Fig. S1a, b). Subjecting *Cb*Fla to on-blot PAL with aminooxy-biotin to assess the presence of sialic acid modification and probing the blot with HRP-streptavidin, we detected biotinylation in *Cb*Fla co-expressed with *Cb*Maf, but not in singly expressed *Cb*Fla (Fig. 1d–f), indicative of *Cb*Maf activity. Further, intact mass analysis confirmed that *Cb*Fla is predominantly present in the modified form when co-expressed with *Cb*Maf, whereas singly expressed CbFla showed a single m/z peak with 130 Da difference from its theoretical mass (30,270 Da), as expected with N-terminal methionine cleavage (Fig. 1g, h and Supplementary Fig. S2a). *Cb*Fla co-expressed with *Cb*Maf showed peaks that indicated robust sialylation and KDOylation, with each modified peak separated by a difference of 291 Da or 220 Da, which corresponds to the molecular weight of Neu5Ac (309 Da) or KDO (238 Da), respectively, after its transfer to protein with loss of water (Fig. 1h). Tandem mass spectrometry of the tryptic digest of *Cb*Fla and analysis of glycosites with a probabilistic approach confirmed these results - *Cb*Fla expressed singly in EV136 cells did not show any modification but *Cb*Fla co-expressed with *Cb*Maf in EV136 cells showed the presence of nine glycosites that included five serine residues (S127, S143, S172, S177, and S199) and four threonine residues (T129, T133, T159, and T197) modified either with Neu5Ac or KDO (Supplementary Data 1, Supplementary Tables S3–S8, and Supplementary Fig. S3a).

### *Cb*Maf might require the structural context provided by D1 but not D0 helices of *Cb*Fla for robust Ser/Thr glycosylation

We designed *Cb*Fla constructs, *Cb*FlaA1, *Cb*FlaA3, and *Cb*FlaA4, to contain the entire D1 loop region but possess varying lengths of the D0/D1 helical region in order to identify the minimal structural scaffold, if any, required by *Cb*Maf for *Cb*Fla glycosylation. As visible from the images of the domain architecture and AphaFold2-predicted three-dimensional structural model, the *Cb*FlaA1 construct comprises only the ND1b helix and the D1 loop region (Fig. 2a, b). *Cb*FlaA3 comprises the ND1b helix, the D1 loop region, and equivalent lengths of the ND1a domain (represented as ND1a3) and CD1 domain (represented as CD1_3), such that a part of the D1 loop containing two anti-parallel beta-strands extends below the ND1 and CD1 helices (Fig. 2f, g). *Cb*FlaA4 comprises the ND1b helix, the D1 loop region and equivalent lengths of the ND1a domain (represented as ND1a2 and ND1a3), and CD1 domain (including CD1_2 and CD1_3) such that the entire D1 loop region can be stabilized/supported by the adjacent ND1 and CD1 helices (Fig. 2k, l).

*Cb*FlaA1, *Cb*FlaA3, and *Cb*FlaA4 constructs were successfully expressed, with or without *Cb*Maf co-expression (Fig. 2c, h, m and Supplementary Fig. S4a–c), in *E. coli* EV136 cells and enriched by Ni-NTA metal ion affinity chromatography (Supplementary Fig. S1c–h). *Cb*FlaA1 and *Cb*FlaA3 did not show any biotinylation upon PAL, indicating no modification, even when co-expressed with *Cb*Maf (Fig. 2c, h). *Cb*FlaA4 co-expressed with *Cb*Maf displayed slower electrophoretic mobility than expected, suggesting that it was modified, and on-blot PAL confirmed the modification of *Cb*FlaA4 by *Cb*Maf (Fig. 2m).

Intact mass analysis confirmed the molecular masses of singly expressed *Cb*FlaA1, *Cb*FlaA3, and *Cb*FlaA4, with 130 Da difference from their theoretical masses (10105.39 Da of *Cb*FlaA1, 14315.11 Da of *Cb*FlaA3, 18075.25 Da of *Cb*FlaA4), indicative of N-terminal methionine cleavage (Fig. 2d, i, n). Upon co-expression with *Cb*Maf, *Cb*FlaA1, and *Cb*FlaA3 each showed two additional peaks of low to moderate intensity, each separated by 291 Da from its previous peak, indicating a low degree of sialylation (in terms of the number of glycosites), which might explain the absence of biotinylation upon PAL (Fig. 2e, j and Supplementary Fig. S2b, c). However, *Cb*FlaA4 co-expressed with *Cb*Maf showed several additional peaks, corresponding to robust modification with seven or eight Neu5Ac moieties and one KDO moiety (Fig. 2o and Supplementary Fig. S2d).

**Fig. 1 | Sialylation of full-length *Cb*Fla by co-expressed *Cb*Maf in *E. coli* EV136 cells cultured in minimal medium. a** Schematic representation of *Cb*Fla modification by *Cb*Maf using CMP-sialic acid and CMP-KDO as donor sugars in *E. coli* EV136 cells. Created with BioRender.com. **b** Domain architecture of full-length *Cb*Fla. ND1a and CD1 have been partitioned into ND1a1, ND1a2, and ND1a3, and CD1_1, CD1_2 and CD1_3, respectively and colored differently, based on the lengths of the ND1b helix and the D1 loop region lying adjacent to ND1a and CD1 in the AlphaFold2-predicted structural model in **c**. **c** AlphaFold2-predicted three-dimensional structural model of full-length *Cb*Fla. D0 is colored golden, D1 helices are colored in different shades of purple, and the D1 loop region is colored gray, similar to the domain architecture in **b**. **d** Western blot analysis of recombinant *Cb*Fla expressed from pET-28a(+) vector in EV136 cells detected with mouse anti-6XHis antibody. **e** The expression of *Cb*Maf was detected with an anti-S tag antibody. **f** On-blot periodate oxidation and aniline-catalyzed ligation with aminooxy biotin of *Cb*Fla singly expressed or co-expressed with *Cb*Maf in EV136 cells. Biotinylation was detected with HRP–streptavidin. Intact mass measurements of *Cb*Fla expressed singly (**g**) or co-expressed with *Cb*Maf (**h**) and purified from EV136 cells. The proteins were acetone precipitated and subjected to LC–MS in positive mode ionization. Insets show the ionization spectrum of *Cb*Fla.

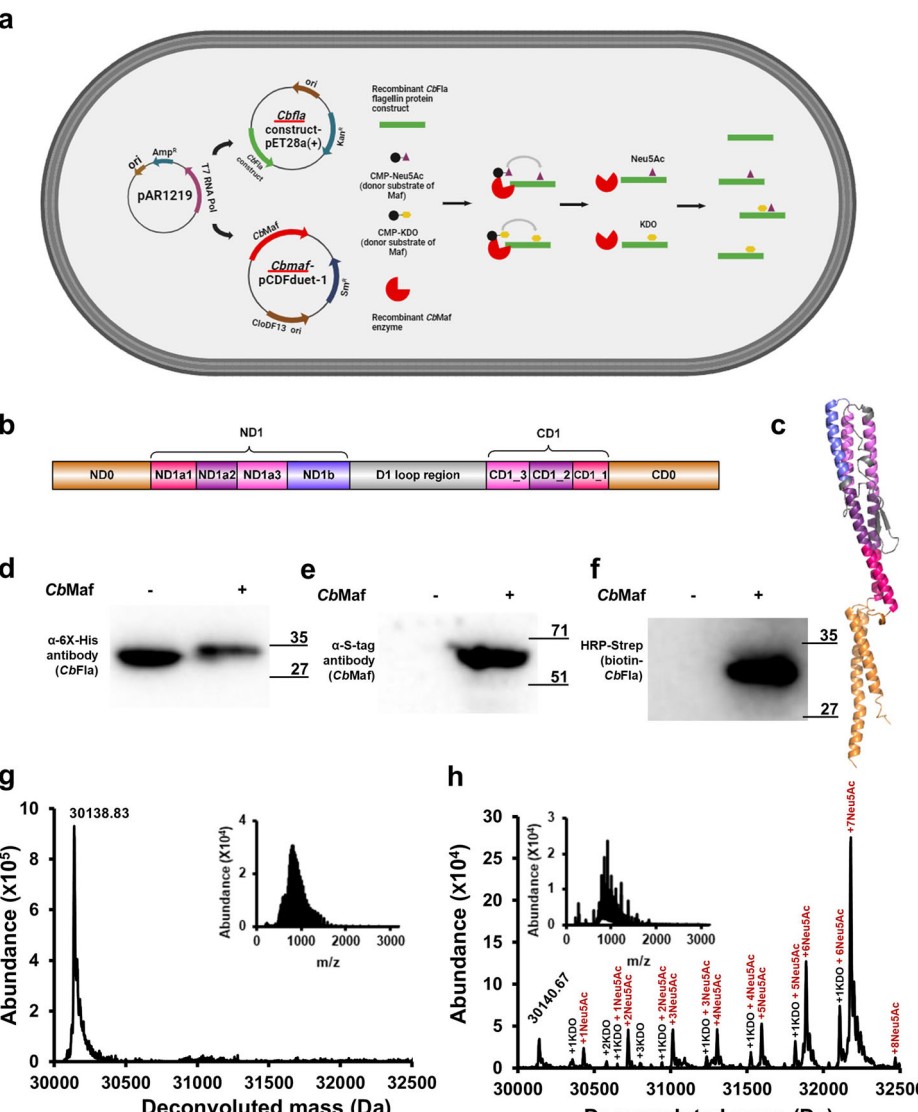

Disregarding the number of glycosites, we observed that almost all the *Cb*FlaA4 protein was glycosylated in three independent protein preparations, similar to full-length *Cb*Fla, indicating that the D0 domain was dispensable for recognition and glycosylation by Maf under the condition of heterologous co-overexpression that we adopted for the study (Fig. 3). A very small fraction of *Cb*FlaA3 was glycosylated (perhaps due to sub-optimal temporal or spatial co-expression with *Cb*Maf or due to sub-optimal folding/stabilization of the D1 loop region in this construct), but a significant fraction of *Cb*FlaA1 was glycosylated, indicating that the short ND1b is sufficient to provide structural support/stability to the D1 loop region and render it conformationally accessible for glycosylation by *Cb*Maf (Fig. 3).

Tandem mass spectrometry of the tryptic digest of *Cb*FlaA1 co-expressed with *Cb*Maf in *E. coli* EV136 cells (Supplementary Data 1, Supplementary Tables S4, S9–S11) did not yield any confidently assigned glycosite (Supplementary Fig. S3b). Tandem mass spectrometry of the tryptic digest of enriched *Cb*FlaA3 co-expressed with *Cb*Maf in *E. coli* EV136 cells (Supplementary Data 1, Supplementary Tables S4, S12–S15) indicated sialylation at S195 and S200 (Supplementary Fig. S3c), and tandem mass spectrometry of the tryptic digest of *Cb*FlaA4 co-expressed with *Cb*Maf in *E. coli* EV136 cells (Supplementary Data 1, Supplementary Tables S4, S16–S19) indicated six glycosites (Supplementary Fig. S3d) - three serine residues (S67@, S112@# and S117@#) and three threonine residues (T99@,

T130@ and T137@) modified with Neu5Ac (@) or KDO (#). Singly expressed *Cb*FlaA1, *Cb*FlaA3, and *Cb*FlaA4 did not show any modification (Supplementary Fig. S3b, c, d). The comparison of the precursor ion intensities of the modified glycopeptides, as generated from the tandem mass spectrometry data of *Cb*FlaA1, *Cb*FlaA3 and *Cb*FlaA4 with *Cb*Fla, co-expressed with *Cb*Maf, indicated that *Cb*FlaA1 and *Cb*FlaA3 were hardly modified, whereas *Cb*FlaA4 was robustly modified in comparison to *Cb*Fla (Supplementary Fig. S3n–p).

## *Cb*Maf glycosylates *Cb*Fla constructs with reduced D1 loop regions and modifies additional non-canonical Ser/Thr residues

Surface exposed Ser/Thr residues in the D1 loop region are modified by *Cb*Maf. We designed *Cb*Fla constructs, *Cb*FlaA5, *Cb*FlaA7, and *Cb*FlaA11, to contain the entire D0 and D1 (ND1a and CD1) helical domains but with deletions of the D1 loop (and ND1b) region in order to assess if a mini loop region would be amenable to robust modification by *Cb*Maf. The helical domains, ND0, CD0, ND1a, and CD1, are in similar fold and orientation to that of full-length *Cb*Fla in all these three constructs, and ND1a and CD1 are linked by a very short six-residue D1 loop region "GKGAKS" in *Cb*FlaA5, by the ND1b helix followed by a 33-residue D1 loop region in *Cb*FlaA7, and by a non-native hexa-glycine loop "GGGGGG" in *Cb*FlaA11 (Fig. 4a, b, f, g, k, l).

*Cb*FlaA5, *Cb*FlaA7, and *Cb*FlaA11 constructs were successfully expressed, with or without *Cb*Maf co-expression (Fig. 4c, h, m and

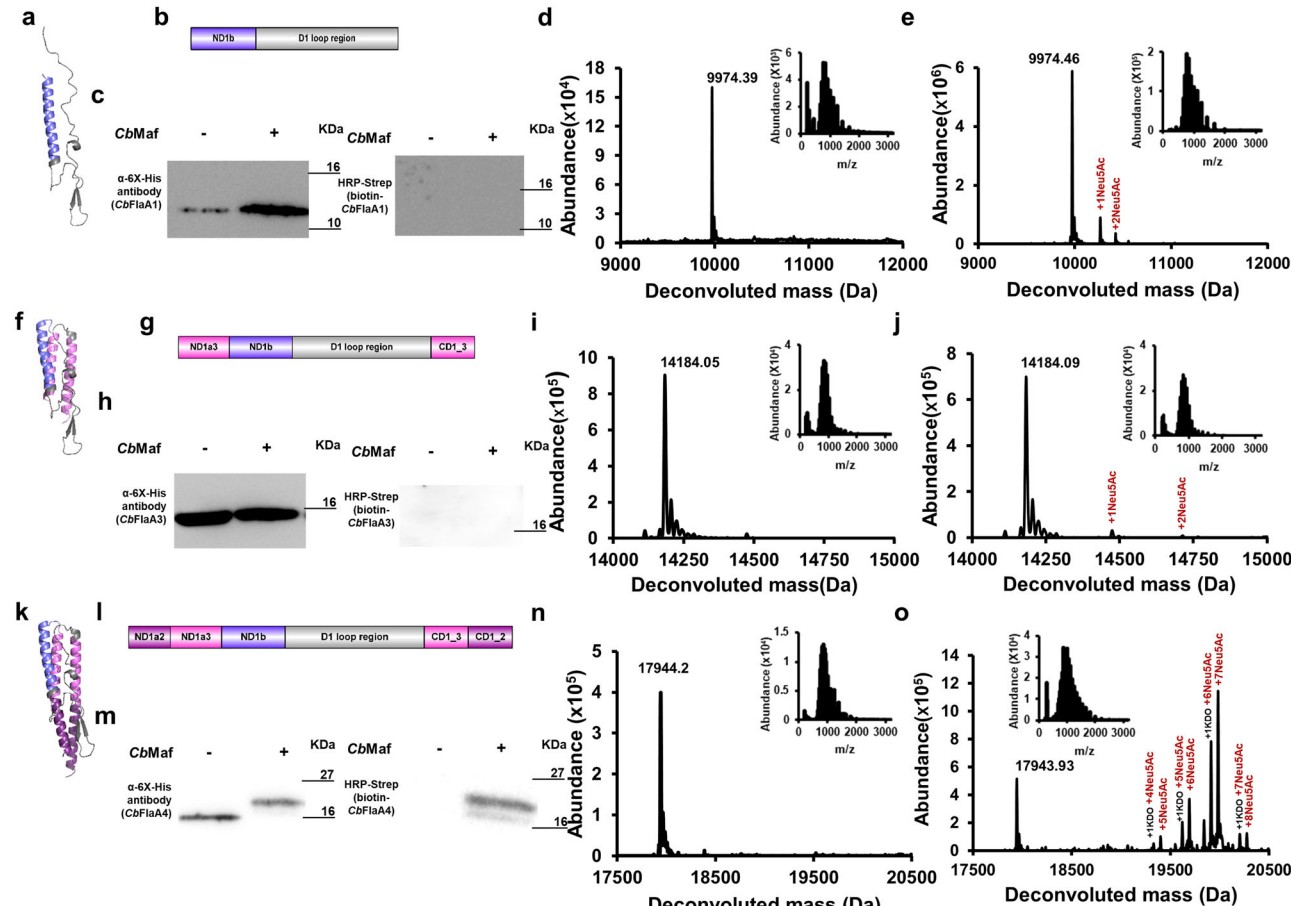

**Fig. 2 | Sialylation of structural determinants of *Cb*Fla by co-expressed *Cb*Maf in *E. coli* EV136 cells cultured in minimal medium.** Domain architectures of *Cb*FlaA1 (**a**), *Cb*FlaA3 (**f**) and *Cb*FlaA4 (**k**). AlphaFold2-predicted structural models of *Cb*FlaA1 (**b**), *Cb*FlaA3 (**g**) and *Cb*FlaA4 (**l**). Western blot analysis of recombinant *Cb*FlaA1 (**c**), *Cb*FlaA3 (**h**) and *Cb*FlaA4 (**m**) singly expressed or co-expressed with *Cb*Maf in EV136 cells. Western analysis was performed with mouse anti-6XHis antibody (to assess protein expression) and with on-blot PAL followed by HRP-streptavidin (to assess protein modification with sialic acid). Intact mass measurements of *Cb*FlaA1, *Cb*FlaA3, and *Cb*FlaA4 expressed singly (**d**, **i**, **n**) or co-expressed with *Cb*Maf (**e**, **j**, **o**) and purified from EV136 cells. The proteins were acetone precipitated and subjected to LC–MS in positive mode ionization. Insets show the ionization spectra of *Cb*FlaA1, *Cb*FlaA3 and *Cb*FlaA4, respectively.

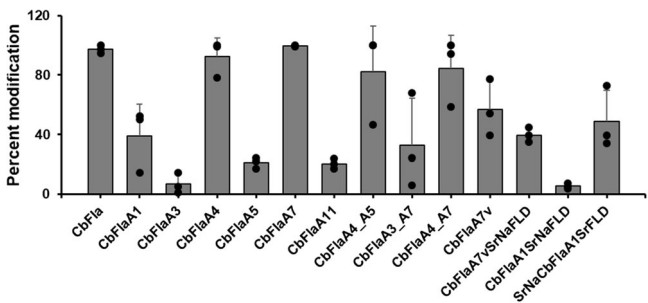

**Fig. 3 | Percent modified fractions of *Cb*Fla constructs when co-expressed with *Cb*Maf in *E. coli* EV136 cells.** Means are plotted as bars and individual data points are included; *n* = 3 biological replicates (three independent protein preparations). Error bars represent standard deviations.

Supplementary Fig. S4d–f) in *E. coli* EV136 cells, and enriched by Ni-NTA metal ion affinity chromatography (Figs. S1i-1n). On-blot PAL indicated robust modification of *Cb*FlaA7, a very low level of modification in *Cb*FlaA5, and none in *Cb*FlaA11, when each was co-expressed with *Cb*Maf (Fig. 4c, h, m).

The intact mass analysis confirmed the molecular masses of singly expressed *Cb*FlaA5, *Cb*FlaA7 and *Cb*FlaA11, with 130 Da difference from their theoretical masses (22650.45 Da of *Cb*FlaA5, 28034.52 Da of *Cb*FlaA7,

22779.44 Da of *Cb*FlaA11), indicative of N-terminal methionine cleavage (Fig. 4d, i, n). Upon co-expression with *Cb*Maf, the intact mass spectra of *Cb*FlaA5 and *Cb*FlaA11 each showed two additional peaks of low intensity, each peak being separated by a mass of 291 Da from its previous peak (Fig. 4e, o and Supplementary Fig. S2e, g), indicative of a low degree of sialylation in these protein constructs, which might explain the low/no biotinylation in PAL. Since the D1 loop region of *Cb*FlaA5 contains only one serine residue and no threonine residue and the D1 loop region of *Cb*FlaA11 lacks any serine or threonine residue, the modification of two sites in these proteins indicates that *Cb*Maf is able to modify non-canonical glycosites, which are perhaps now surface-exposed in the absence of the complete D1 loop region. *Cb*FlaA7, when co-expressed with *Cb*Maf, yielded three peaks with masses that indicated robust modification with up to five Neu5Ac moieties and one KDO moiety (Fig. 4j and Supplementary Fig. S2f). Comparing the overall percent protein modification of three independent protein preparations, we found that whereas only a small fraction of protein was glycosylated in *Cb*FlaA5 and *Cb*FlaA11, almost all the protein was glycosylated in *Cb*FlaA7, similar to full-length *Cb*Fla, indicating that the removal of a significant chunk of the D1 loop region (i.e., a 23-residue region including two Ser residues) and retention of just the 33-amino acid region did not affect glycosylation of the remaining Ser/Thr residues in the D1 loop region (Fig. 3).

Analysis of the tandem mass spectra of the tryptic digest of *Cb*FlaA5 (Supplementary Data 1, Supplementary Tables S4, S20–S25), co-expressed with *Cb*Maf in *E. coli* EV136 cells, indicated nine confidently assigned

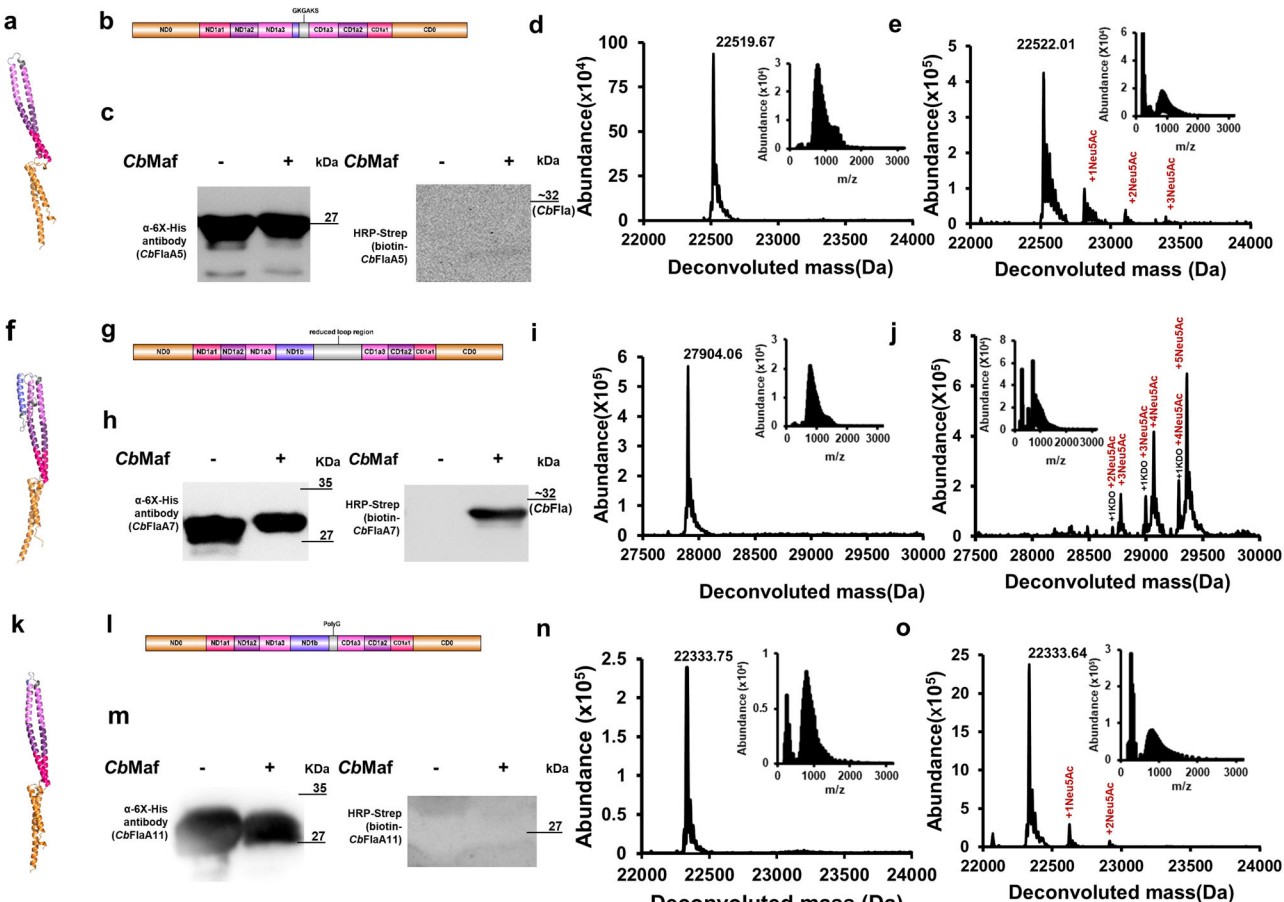

**Fig. 4 | Sialylation of sequence determinants of *Cb*Fla by co-expressed *Cb*Maf in *E. coli* EV136 cells cultured in minimal medium.** Domain architectures of *Cb*FlaA5 (**a**), *Cb*FlaA7 (**f**) and *Cb*FlaA11 (**k**). AlphaFold2-predicted structural models of *Cb*FlaA5 (**b**), *Cb*FlaA7 (**g**) and *Cb*FlaA11 (**l**). Western blot analysis of recombinant *Cb*FlaA5 (**c**), *Cb*FlaA7 (**h**) and *Cb*FlaA11 (**m**) singly expressed or co-expressed with *Cb*Maf in EV136 cells. Western analysis was performed with mouse anti-6XHis antibody (to assess protein expression) and with on-blot PAL followed by HRP-streptavidin (to assess protein modification with sialic acid). For the HRP-streptavidin blots of (**c**) and (**h**), only one size marker (*Cb*Fla) is available as the visible light images with the pre-stained molecular marker were inadvertently not saved; the molecular mass range of the proteins on the blot may be approximated from the anti-6XHis gels run in parallel. Intact mass measurements of *Cb*FlaA5, *Cb*FlaA7, and *Cb*FlaA11 expressed singly (**d**, **i**, **n**) or co-expressed with *Cb*Maf (**e**, **j**, **o**) and purified from EV136 cells. The proteins were acetone precipitated and subjected to LC–MS in positive mode ionization. Insets show the ionization spectra of *Cb*FlaA5, *Cb*FlaA7 and *Cb*FlaA11, respectively.

glycosites - six serine residues (S50@, S64@, S67@, S72@, S98@# and S110@) and three threonine residues (T76@, T117@, and T124@) modified with Neu5Ac (@) or KDO (#), albeit one glycosite, T124#, was assigned in singly expressed *Cb*FlaA5, too, using the same parameters (Supplementary Fig. S3e). Manual inspection of the latter did not indicate the presence of glycan-modified peptide ions confirming glycosylation in singly expressed *Cb*FlaA5 (Supplementary Fig. S3e). Analysis of the tandem mass spectra of the Glu-C digest of *Cb*FlaA5 (Supplementary Data 1, Tables S4, S20–S25), co-expressed with *Cb*Maf, indicated sialylation at S64 and S67, and no glycosite assignment was made in Glu-C digested, singly expressed *Cb*FlaA5 (Supplementary Fig. S3e). Analysis of the tandem mass spectra of the tryptic digest of *Cb*FlaA7 (Supplementary Data 1, Tables S4, S26–S30), co-expressed with *Cb*Maf in *E. coli* EV136 cells, yielded the confident assignment of eleven glycosites - seven serine residues (S50@, S127@, S142@, S148@#, S153@, S159@ and S175@) and four threonine residues (T129@, T141@, T166@ and T173@) modified with Neu5Ac (@) or KDO (#), and singly expressed *Cb*FlaA7 did not show any modification (Supplementary Fig. S3f). No glycosite could be assigned in Glu-C digested *Cb*FlaA7 (Supplementary Data 1, Supplementary Tables S4, S26–S30), whether expressed singly or co-expressed with *Cb*Maf (Supplementary Fig. S3f). We are not certain why fewer glycosylation sites were assigned using Glu-C; considering the better results with trypsin digest, we proceeded further with only tryptic digests for other constructs analyzed by MS/MS in this study.

A comparison of the precursor ion intensities of the modified glycopeptides (obtained from the tandem mass spectrometry data) indicated slightly lower percent modification around the "GAKSIS" loop in *Cb*FlaA5 than in *Cb*Fla, higher percent modification for the reduced D1 loop region in *Cb*FlaA7 than in *Cb*Fla, and similarly ranging high percent modification values for some of the new glycosites observed in *Cb*FlaA5 and *Cb*FlaA7 when co-expressed with *Cb*Maf (Supplementary Fig. S3q, r). This supported our conclusion that *Cb*Maf modifies additional non-canonical glycosites in the absence of the complete D1 loop region.

## *Cb*Maf glycosylates *Cb*Fla constructs that contain truncated D1 helices and reduced D1 loops

Considering the results obtained with the D0/D1 deletion constructs and D1 loop deletion constructs, we designed five additional *Cb*Fla constructs. *Cb*FlaA4_A5, *Cb*FlaA3_A7, and *Cb*FlaA4_A7 combine the truncated D1 helical and reduced D1 loop region elements from our aforementioned constructs (Fig. 5a, b, f, g, k, l). *Cb*FlaA7v comprises only the ND1b part of the D1 helical domain and the reduced D1 loop region of *Cb*FlaA7 (Fig. 5p), the structure of which is disordered or not predicted with confidence by AlphaFold2 (Fig. 5q). *Cb*FlaD0A7vloop comprises just the D0 helical domain and the reduced D1 loop region of *Cb*FlaA7 (Supplementary Fig. S5a), and the AlphaFold2-predicted structural model indicates that the

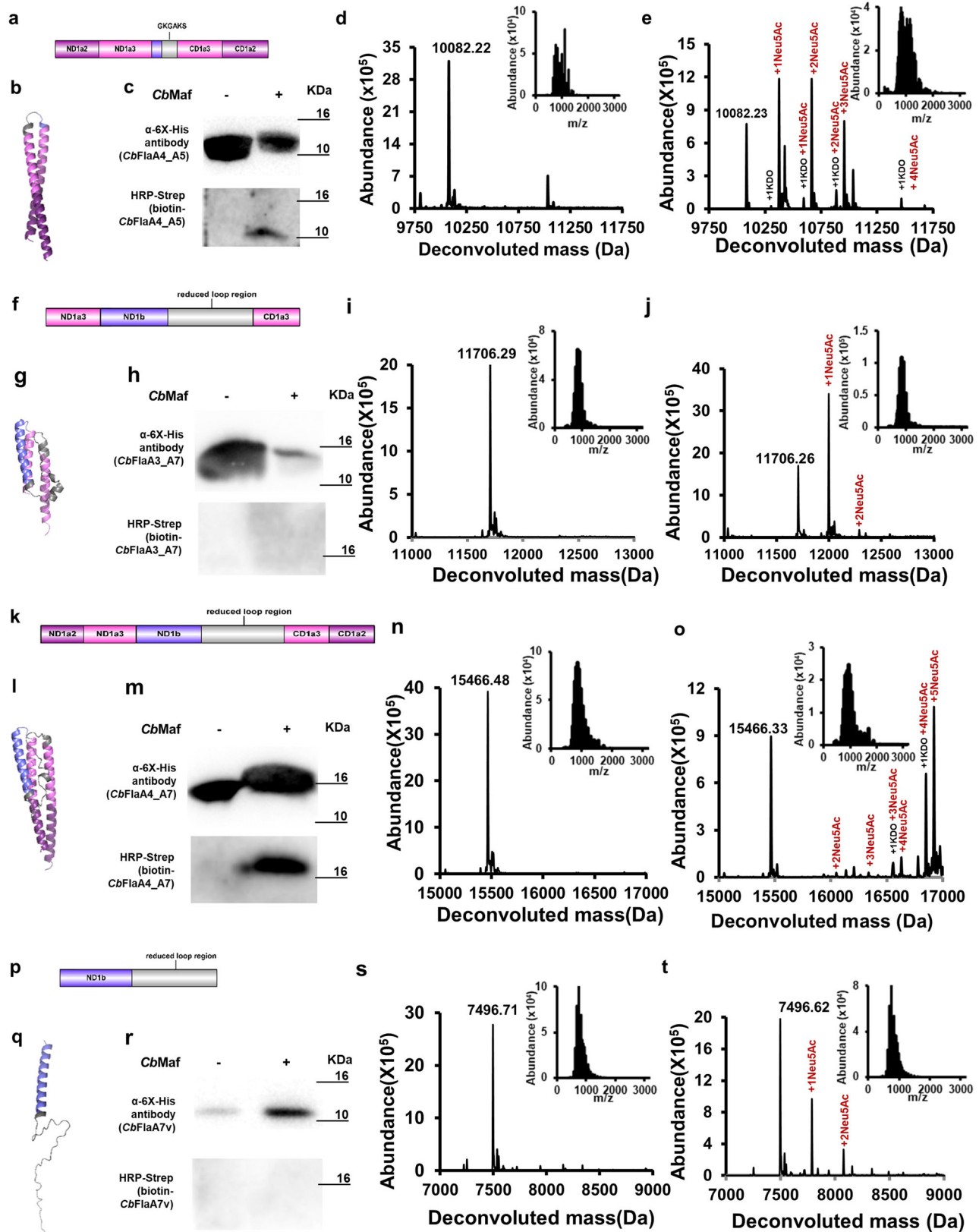

**Fig. 5 | Sialylation of structural and sequence determinants of *Cb*Fla by co-expressed *Cb*Maf in *E. coli* EV136 cells cultured in minimal medium.** Domain architectures of *Cb*FlaA4_A5 (**a**), *Cb*FlaA3_A7 (**f**), *Cb*FlaA4_A7 (**k**) and *Cb*FlaA7v (**p**). AlphaFold2-predicted structural models of *Cb*FlaA4_A5 (**b**), *Cb*FlaA3_A7 (**g**), *Cb*FlaA4_A7 (**l**) and *Cb*FlaA7v (**q**). Western blot analysis of recombinant *Cb*FlaA4_A5 (**c**), *Cb*FlaA3_A7 (**h**), *Cb*FlaA4_A7 (**m**) and *Cb*FlaA7v (**r**) singly expressed or co-expressed with *Cb*Maf in EV136 cells. Western analysis was

performed with mouse anti-6XHis antibody (to assess protein expression) and with on-blot PAL followed by HRP-streptavidin (to assess protein modification with sialic acid). Intact mass measurements of *Cb*FlaA4_A5, *Cb*FlaA3_A7, *Cb*FlaA4_A7 and *Cb*FlaA7v expressed singly (**d**, **i**, **n**, **s**) or co-expressed with *Cb*Maf (**e**, **j**, **o**, **t**) and purified from EV136 cells. The proteins were acetone precipitated and subjected to LC–MS in positive mode ionization. Insets show the ionization spectra of *Cb*FlaA4_A5, *Cb*FlaA3_A7, *Cb*FlaA4_A7 and *Cb*FlaA7v, respectively.

region corresponding to the reduced loop region is instead a helix in this construct (Supplementary Fig. S5b).

All the five constructs were successfully expressed, with or without *Cb*Maf co-expression, in *E. coli* EV136 cells (Fig. 5c, h, m, r and Supplementary Figs. S4g–k, S5c), and enriched by Ni-NTA metal ion affinity chromatography (Supplementary Fig. S1o–x). On-blot PAL with aminooxy biotin indicated that *Cb*FlaA4_A5 and *Cb*FlaA4_A7, but not *Cb*FlaA3_A7, *Cb*FlaA7v and *Cb*FlaD0A7loop, were robustly modified when co-expressed with *Cb*Maf (Fig. 5c, h, m, r, and Supplementary Fig. S5c).

The intact mass analysis confirmed the molecular masses of singly expressed *Cb*FlaA4_A5, *Cb*FlaA3_A7, *Cb*FlaA4_A7, *Cb*FlaA7v and *Cb*FlaD0A7loop with 130 Da difference from their theoretical masses (10213.30 Da of *Cb*FlaA4_A5, 11837.23 Da of *Cb*FlaA3_A7, 15597.37 Da of *Cb*FlaA4_A7, 7627.51 Da of *Cb*FlaA7v and 13406.28 Da of *Cb*FlaD0A7loop), indicative of N-terminal methionine cleavage (Fig. 5d, i, n, s and Supplementary Fig. S5d). *Cb*FlaA4_A5 co-expressed with *Cb*Maf showed additional peaks of masses that indicated robust modification with up to four Neu5Ac moieties and one KDO moiety, albeit no glycosites could be predicted by analysis of tandem mass spectrometry data of the tryptic digest of *Cb*FlaA4_A5 (Fig. 5e, Supplementary Figs. S2h, S3g, and Supplementary Data 1, Supplementary Tables S4, S31, S32). *Cb*FlaA4_A7 co-expressed with *Cb*Maf showed additional peaks with masses indicating robust modification with up to five Neu5Ac moieties and one KDO moiety (Fig. 5o and Supplementary Fig. S2j). The intact mass spectrum of *Cb*FlaA3_A7 co-expressed with *Cb*Maf showed two additional peaks with masses that indicated modification with two Neu5Ac moieties (Fig. 5j and Supplementary Fig. S2i). *Cb*FlaA7v co-expressed with *Cb*Maf showed two additional peaks of lower intensity, each being separated by 291 Da from its previous peak, indicating modification with up to two Neu5Ac moieties (Fig. 5t and Supplementary Fig. S2k). The lower degree of modification in *Cb*FlaA3_A7 and *Cb*FlaA7v might explain the absence of biotinylation upon PAL labeling. The intact mass spectrum of *Cb*FlaD0A7loop co-expressed with *Cb*Maf did not show any additional modified peak (Supplementary Fig. S5e), in agreement with the PAL labeling result, indicating that *Cb*FlaD0A7vloop was not sialylated by *Cb*Maf, likely because the reduced D1 loop region adopts a helical structure instead in this protein construct. Tandem mass spectrometry of the tryptic digest of *Cb*FlaDoA7loop, co-expressed with *Cb*Maf, also did not indicate any modification (Supplementary Fig. S3h and Supplementary Data 1, Supplementary Tables S4, S33, S34).

We compared the overall percent protein modification of three independent protein preparations and found that most of the protein was modified in *Cb*FlaA4_A5 and *Cb*FlaA4_A7 constructs, similar to *Cb*FlaA7, *Cb*FlaA4, and *Cb*Fla (Fig. 3). This implied that a flagellin acceptor substrate lacking the D0 domain and containing a reduced D1 loop region and truncated D1 helical domain is sufficient for recognition and sialylation by *Cb*Maf. We found that although *Cb*FlaA3_A7 displayed a lower percent protein modification than *Cb*FlaA4_A7, it was modified more than *Cb*FlaA3. This validated our reasoning that the low percent protein modification in *Cb*FlaA3 (Figs. 2j and 3) was probably due to the overhanging D1 loop region lacking structural support/stabilization from the ND1 and CD1 helices. Importantly, we observed modification of a considerable fraction of the shortest construct, *Cb*FlaA7v (69 residues long; includes 59 residues from the ND1b and D1 loop region), containing only the ND1b helix and a reduced D1 loop region, indicating that this was also a suitable mini-flagellin acceptor substrate for *Cb*Maf (Fig. 3).

### *Cb*Maf glycosylates mini-flagellin protein chimeras
Having determined that *Cb*Maf could glycosylate mini-flagellin protein constructs, we assessed if *Cb*Maf could also glycosylate mini-flagellin protein chimeras. We selected an unrelated protein, *Sr*NaFLD[33] from *Streptosporangium roseum*, with two discrete beta-sandwich domains – Na (Non-Putative Carbohydrate-Binding Module-associated domain; function unknown) and FLD (F-type lectin domain; L-fucose binding) (Fig. 6a, b) for making the chimeras. We recombinantly expressed *Sr*NaFLD in *E. coli* EV136 cells with or without *Cb*Maf co-expression (Fig. 6c and

Supplementary Fig. S4l) and partially purified it using Ni-NTA (Supplementary Fig. S1y, z). On-blot PAL indicated no modification in *Sr*NaFLD, when co-expressed with *Cb*Maf (Fig. 6c). Intact mass analysis of *Sr*NaFLD (theoretical molecular mass: 27982.47 Da), with or without *Cb*Maf co-expression, yielded a single peak corresponding to the molecular mass of *Sr*NaFLD expected with N-terminal methionine loss and formation of a disulfide bond (Fig. 6d, e). Tandem mass spectrometry of the tryptic digest of *Sr*NaFLD, co-expressed with *Cb*Maf in *E. coli* EV136 cells, did not yield any modified peptides with Modscore ≥19 (Supplementary Fig. S3i and Supplementary Data 1, Supplementary Tables S4, S35, S36).

We made five mini-flagellin protein chimeras, four (*Cb*FlaA1_*Sr*NaFLD, *Cb*FlaA7v_*Sr*NaFLD, GKGAKS_*Sr*NaFLD, and *Cb*Fla_A1helix_GKGAKS_*Sr*NaFLD) with the mini-flagellin region at the N-terminus, and one (*Sr*Na_*Cb*FlaA1_*Sr*FLD) with the mini-flagellin region sandwiched in between Na and FLD (Fig. 6f, k and Supplementary Fig. S5f, k, p). We used four mini-flagellin regions, a "GKGAKS" hexapeptide from the D1 loop region, a 32-residue "A1helix_GKGAKS" region comprising the ND1b helix, and the hexapeptide "GKGAKS", the 59-residue "*Cb*FlaA7v" region comprising the ND1b helix and the reduced D1 loop region of the *Cb*FlaA7v construct, and the 83-residue "*Cb*FlaA1" region comprising the ND1b helix and the complete D1 loop region of the *Cb*FlaA1 construct (Fig. 6g, l and Supplementary Fig. S5g, l, q). We recombinantly expressed the mini-flagellin protein chimeras, with or without *Cb*Maf, in *E. coli* EV136 (Fig. 6h, m and Supplementary Figs. S4m–q, S5h, m, r) and purified them (Supplementary Fig. S1aa-aj).

On-blot PAL indicated that *Cb*FlaA7v_*Sr*NaFLD and *Sr*Na_*Cb*FlaA1_*Sr*FLD but not *Cb*FlaA1_*Sr*NaFLD, GKGAKS_*Sr*NaFLD and *Cb*Fla_A1helix_GKGAKS_*Sr*NaFLD, were modified with sialic acids, when co-expressed with *Cb*Maf (Fig. 6h, m and Supplementary Fig. S5h, m and r). Intact mass analysis of singly expressed *Cb*FlaA7v_*Sr*NaFLD, *Sr*Na_*Cb*FlaA1_*Sr*FLD, *Cb*FlaA1_*Sr*NaFLD, GKGAKS_*Sr*NaFLD, and *Cb*Fla_A1helix_GKGAKS_*Sr*NaFLD (theoretical masses: 24526.85 Da, 36745.60 Da, 37004.72 Da, 28454.03 Da, and 31461.34 Da, respectively) yielded single peaks corresponding to masses expected with N-terminal methionine cleavage and formation of a disulfide bond (Fig. 6i, n and Supplementary Fig. S5i, n, s). Importantly, abundant additional peaks were observed in three independent protein preparations of *Cb*FlaA7v_*Sr*NaFLD and *Sr*Na_*Cb*FlaA1_*Sr*FLD, when co-expressed with *Cb*Maf, indicating robust modification with up to three Neu5Ac moieties (Fig. 6j, o and Supplementary Fig. S2l, m). *Cb*FlaA1_*Sr*NaFLD co-expressed with *Cb*Maf displayed one additional peak of low intensity separated by 291 Da, corresponding to modification with one Neu5Ac moiety (Supplementary Fig. S5j and Supplementary Fig. S2n). The lower degree of modification in *Cb*FlaA1_*Sr*NaFLD co-expressed with *Cb*Maf might explain the absence of biotinylation upon PAL labeling. No modified peaks were observed in GKGAKS_*Sr*NaFLD and *Cb*Fla_A1helix_GKGAKS_*Sr*NaFLD, when co-expressed with *Cb*Maf (Supplementary Fig. S5o, t), in agreement with the PAL labeling result, confirming that GKGAKS_*Sr*NaFLD and *Cb*Fla_A1helix_GKGAKS_*Sr*NaFLD were not suitable acceptor substrates for *Cb*Maf.

Using tandem mass spectrometry of the tryptic digest of *Sr*Na_*Cb*FlaA1_*Sr*FLD (Supplementary Data 1, Supplementary Tables S4, S37–S40), co-expressed with *Cb*Maf, we identified four Ser glycosites (Supplementary Fig. S3j), three of which are located in the D1 loop region. The remaining Ser residue is present in a flexible loop of the *Sr*FLD, as predicted by the AlphaFold2 software. A comparison of the precursor ion intensities of the modified glycopeptides, as generated from the tandem mass spectrometry data of *Sr*Na_*Cb*FlaA1_*Sr*FLD and full-length *Cb*Fla, both co-expressed with *Cb*Maf, indicated lower albeit significant modification in one peptide in the mini-flagellin chimeric construct. No modified peptides were identified in *Sr*Na_*Cb*FlaA1_*Sr*FLD expressed singly in EV136 cells (Supplementary Fig. S3s). We also performed tandem mass spectrometry of the tryptic digests of GKGAKS_*Sr*NaFLD and *Cb*Fla_A1helix_GKGAKS_*Sr*NaFLD, co-expressed with *Cb*Maf in *E. coli* EV136 cells, and identified no modified peptides (Supplementary Fig. S3k, l and Supplementary Data 1, Supplementary Tables S4, S41–S44).

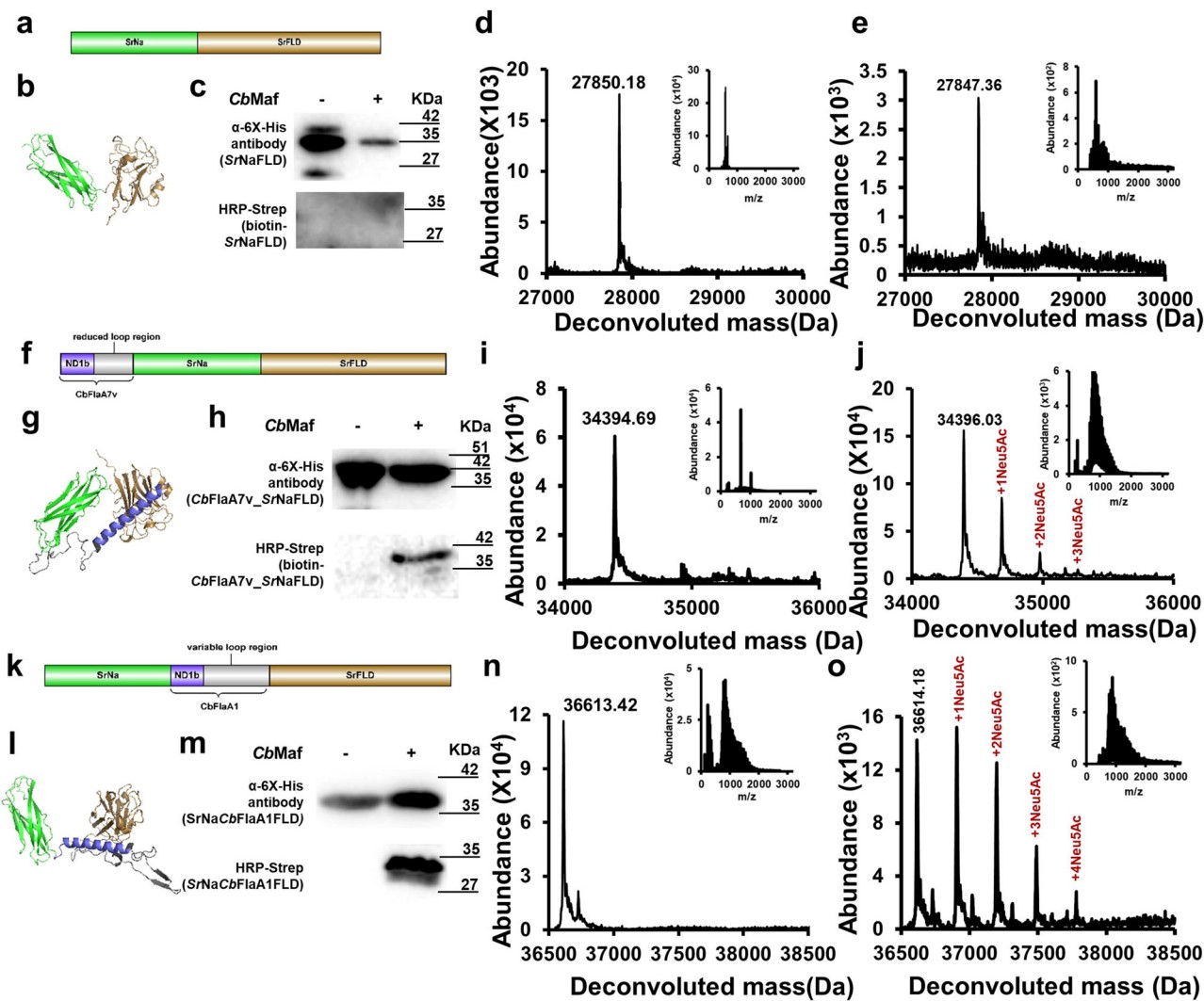

**Fig. 6 | Sialylation of recombinant chimeras of *Cb*Fla deletions with *Sr*NaFLD by co-expressed *Cb*Maf in *E. coli* EV136 cells cultured in minimal medium.** Domain architectures of *Sr*NaFLD (**a**), *Cb*FlaA7v-*Sr*NaFLD (**f**) and *Sr*Na*Cb*FlaA1*Sr*NaFLD (**k**). AlphaFold2-predicted structural models of *Sr*NaFLD (**b**), *Cb*FlaA7v-*Sr*NaFLD (**g**) and *Sr*Na*Cb*FlaA1*Sr*NaFLD (**l**). Western blot analysis of recombinant *Sr*NaFLD (**c**), *Cb*FlaA7v-*Sr*NaFLD (**h**) and *Sr*Na*Cb*FlaA1*Sr*NaFLD (**m**) singly expressed or co-expressed with *Cb*Maf in EV136 cells. Western analysis was performed with mouse

anti-6XHis antibody (to assess protein expression) and with on-blot PAL followed by HRP-streptavidin (to assess protein modification with sialic acid). Intact mass measurements of *Sr*NaFLD, *Cb*FlaA7v-*Sr*NaFLD and *Sr*Na*Cb*FlaA1*Sr*NaFLD expressed singly (**d**, **i**, **n**) or co-expressed with *Cb*Maf (**e**, **j**, **o**) and purified from EV136 cells. The proteins were acetone precipitated and subjected to LC–MS in positive mode ionization. Insets show the ionization spectra of *Sr*NaFLD, *Cb*FlaA7v-*Sr*NaFLD and *Sr*Na*Cb*FlaA1*Sr*NaFLD, respectively.

The results we obtained with the mini-flagellin protein chimeras hint at the requirement of a certain level of D1 loop region stabilization and/or accessibility for *Cb*Maf to modify it. Considering that the mini-flagellin construct, *Cb*FlaA7v, itself showed only two modifications of low abundance (Fig. 5t), the observation of robust modification on *Cb*FlaA7v_*Sr*NaFLD suggests that *Sr*NaFLD provides structural support and stability to the loop region of *Cb*FlaA7v. A higher proportion of modified protein and an increased number of Neu5Ac moieties modifying the protein were observed with *Sr*Na_*Cb*FlaA1_*Sr*FLD as compared to the mini-flagellin construct, *Cb*FlaA1 (Fig. 2e), perhaps due to a stabilizing effect on the *Cb*Fla loop region by the secondary scaffold of *Sr*NaFLD, or due to interactions between the chimeric protein and *Cb*Maf. On the other hand, *Cb*FlaA1_*Sr*NaFLD displayed weaker modification than *Cb*FlaA1, when co-expressed with *Cb*Maf.

### Maf glycosyltransferases do not glycosylate non-cognate flagellins

We further explored the acceptor substrate promiscuity of Maf glycosyl-transferases by co-expressing Maf with non-cognate flagellin. We co-expressed the flagellin proteins of *C. botulinum* (*Cb*Fla) and *G. kaustophilus*

(*Gk*FlaA1 and *Gk*FlaA2) with the Maf proteins of *G. kaustophilus* (*Gk*Maf) and *C. botulinum* (*Cb*Maf), respectively, (Supplementary Fig. S6a–d) in *E. coli* EV136 cells, and partially purified the flagellins by Ni-NTA affinity chromatography (Supplementary Figs. S1ak–am, S6e). We observed no biotinylation upon on-blot PAL labeling, indicative of no modification (Supplementary Fig. S6f).

The intact mass measurement of *Gk*FlaA1 (theoretical mass of 65107.68 Da) co-expressed with *Cb*Maf could not be obtained. However, the intact mass measurements of *Cb*Fla (Supplementary Fig. S6g) and *Gk*FlaA2 (Supplementary Fig. S6h) co-expressed with *Gk*Maf and *Cb*Maf, respectively, showed single peaks corresponding to the molecular masses of the unmodified flagellins with a difference of 130 Da (30138.90 Da for *Cb*Fla and 32927.72 Da for *Gk*FlaA2) from their theoretical masses (30270.12 Da of *Cb*Fla and 33057.84 Da of *Gk*FlaA2) as expected due to N-terminal methionine cleavage. This implied that *Cb*Maf and *Gk*Maf do not glyco-sylate non-cognate flagellins.

We also obtained tandem mass spectrometry data of *Cb*Fla co-expressed with *Gk*Maf. This data, in opposition to the intact mass spec-trometry data, indicated the presence of four glycosites with a Modscore of

≥19. The four sites included two serine residues in the D1 loop region, one serine residue in the ND0 domain, and one threonine residue in the ND1a3 region of the D1 domain (Supplementary Fig. S3m, and Supplementary Data 1, Tables S4, S45–S47). A comparison of the precursor ion intensities of the modified glycopeptides, as generated from the tandem mass spectrometry data of *Cb*Fla, co-expressed with *Cb*Maf and *Gk*Maf, respectively, however, indicated that only one short, tryptic peptide, "K.LSSGLR.I", of *Cb*Fla co-expressed with *Gk*Maf was actually well-modified (Supplementary Fig. S3t). It is important to note here that the same peptide was also predicted to be highly modified in *Gk*FlaA1 and *Gk*FlaA2 by *Gk*Maf, and we had cautioned that this might be an artifact of recombinant overexpression in a heterologous system[18]. This peptide occurs in the N-terminal D0 spoke region (the short loop that is found between the N-terminal D0 and D1 helices), and we noted then that while the frequency of modification (as predicted in all modified peptides by Modscore) was very high in certain datasets, this peptide was completely absent in Asp-N endoproteinase digests of *Gk*FlaA1 and in the Glu-C protease digests of *Gk*FlaA2. Considering this background and the complete absence of modification in the PAL labeling and the intact mass analysis, we conclude that Maf glycosyltransferase is not promiscuous towards non-cognate flagellin substrates.

## Discussion

Protein glycosylation is the most common and widely distributed post-translational modification found in both prokaryotes and eukaryotes[34–36]. Glycosylation has been shown to improve protein folding, solubility, stability, and binding affinity[37,38]. Glycosylation also improves serum half-life by preventing the clearance of the glycoprotein by lectin receptors, and shielding proteolysis-susceptible regions and immunogenic epitopes, and is therefore a particularly important consideration for glycoprotein biologics[39]. Glycosylation, especially with glycans containing negatively charged, acidic residues such as sialic acids, can also alter the physiochemical attributes of proteins and increase water retention. Strategic modification of a protein directly with sialic acids by protein nonulosonic acid transferases such as Maf and FlmG glycosyltransferases might thus impart similar physicochemical attributes and benefits such as protection from proteolysis, immune recognition, and clearance by lectin receptors.

Glycoengineering applications require glycosyltransferases with defined acceptor substrate features. No conserved sequons for glycosylation are known in the flagellin substrates of the Maf glycosyltransferase family. However, the modified residues on flagellin are reported to be found around hydrophobic amino acids or low-complexity regions[40–42], and site-directed mutagenesis of the hydrophobic amino acids around the Ser/Thr glycosites in the D2/D3 domain of *Aeromonas caviae* flagellin leads to detrimental effects on motility[28]. We, therefore, aimed, via this study, to identify the minimal sequence / structural regions of *Cb*Fla required for *Cb*Maf recognition and glycosyltransferase activity and then explore the potential of *Cb*Maf glycosyltransferase in a glycoengineering application. We found that mini-flagellin constructs with truncated helices and/or reduced D1 loops - *Cb*FlaA4, *Cb*FlaA7, *Cb*FlaA4_A7, and *Cb*FlaA4_A5 were glycosylated as efficaciously as full-length *Cb*Fla by *Cb*Maf, and the 69-residue *Cb*FlaA7v was the shortest construct robustly glycosylated by *Cb*Maf.

Our comparison of the glycosylation among the various mini-flagellin constructs provided some indications regarding the acceptor substrate features recognized by *Cb*Maf.

One, the D0 domain is dispensable for recognition and/or modification of flagellin by *Cb*Maf under the conditions of heterologous overexpression of *Cb*Maf and *Cb*Fla adopted in this study. This conclusion is based on the observation that the constructs *Cb*FlaA4 and *Cb*FlaA7, which lack the D0 domain and part of the D1 helical domain, displayed similar glycosylation as *Cb*Fla. It remains to be seen whether the D0 domain is also dispensable for glycosylation of *Cb*Fla by *Cb*Maf under physiological conditions in *C. botulinum*.

Two, the ND1a and CD1 helices might not have any role other than offering structural stability to the D1 loop region. This conclusion is based on the observation that *Cb*Maf glycosylated the constructs, *Cb*FlaA1 and

*Cb*FlaA7v. These mini-flagellin constructs contain only the ND1b domain followed by the loop region (the entire D1 loop region in the case of *Cb*FlaA1 and the reduced D1 loop region of *Cb*FlaA7 in the case of *Cb*FlaA7v), and this suggests that *Cb*Maf might not require the presence of D1 or D0 domains for *Cb*Fla glycosylation. Further, *Cb*Maf glycosylated the mini-flagellin chimeras, *Cb*FlaA1_*Sr*NaFLD, *Cb*FlaA7v_*Sr*NaFLD, and *Sr*Na_*Cb*FlaA1_*Sr*FLD, too, albeit to different extents. This exciting result underscores the dispensability of any specific secondary structural element for *Cb*Maf recognition of the acceptor substrate while hinting at the importance of the correct positioning of the Ser/Thr loop region as the determinant for glycosylation mediated by *Cb*Maf. A limitation of our study is that we did not assess the glycosylation of mini-flagellin constructs lacking the ND1b helix by *Cb*Maf. It is, therefore, not clear from our experiments whether *Cb*Maf requires the ND1b helix as a minimal structural scaffold for appropriate presentation or structural stabilization of the D1 loop Ser/Thr residues.

Three, *Cb*Maf likely recognizes surface-exposed Ser/Thr loop residues that are stabilized in a favorable conformation by contacts within the loop and/or between the loop and the secondary structural elements of the protein. This conclusion is based on the observations that *Cb*Maf does not glycosylate the non-cognate flagellin, *Gk*FlaA2, or the cognate mini-flagellin construct, *Cb*FlaD0A7loop, in which the D1 region adopts a helical conformation rather than a flexible loop conformation as per AlphaFold2 prediction but glycosylates the cognate mini-flagellin constructs and chimera with reduced D1 loop regions, *Cb*FlaA7, *Cb*FlaA5, *Cb*FlaA3_A7, *Cb*FlaA4_A7, *Cb*FlaA4_A5, and *Cb*FlaA7v*Sr*NaFLD, and glycosylates additional Ser/Thr residues in *Cb*FlaA7 and *Cb*FlaA5 as well as in *Cb*FlaA11 (which has a synthetic loop connecting the ND1a and CD1 helices). The glycosylation of additional Ser/Thr residues in *Cb*FlaA5, *Cb*FlaA7, and *Cb*FlaA11 might be due to reduced/altered steric constraints with respect to *Cb*Maf contacting *Cb*Fla and/or increased surface/exposure or access to these residues upon D1 loop truncation. Further, *Cb*FlaA3, which does not possess the ND1a1, ND1a2, CD1_1 and CD1_2 regions, lacks the stabilizing polar contacts between the loop (Q147 and N151) and the ND1b helix (N66 and Q56), which are present in full-length *Cb*Fla, and is subsequently less modified by *Cb*Maf. *Cb*FlaA1, by comparison, retains the D1 loop region in a near-native conformation (as per AlphaFold2 prediction), and is more abundantly glycosylated than *Cb*FlaA3. Also, the very short D1 loop region in *Cb*FlaA4_A5 is robustly glycosylated but the same loop region in *Cb*FlaA5 is glycosylated to a lesser extent and that the mini-flagellin chimeras with similarly short loops, GKGAKS_*Sr*NaFLD and *Cb*FlaA1helix_GKGAKS_*Sr*NaFLD are not glycosylated at all. This suggests that the structural conformation adopted by the Ser/Thr residues might be different in these different contexts and might be critical for recognition by *Cb*Maf. We cannot however rule out the possibility that these recombinant proteins differ in their spatial/temporal co-expression profiles with *Cb*Maf. In this context, we would like to note the higher degree of modification we observed in the full-length *Cb*Fla co-expressed with *Cb*Maf in this study, as compared to our previous study, which used different expression vectors and expression conditions[18].

Our study used intact mass analysis with three independent preparations of purified protein samples to discern the degree of glycosylation. While we did also perform precursor ion intensity analysis on tandem mass spectrometry data, this only provided information on the degree of modification of tryptic peptides, which frequently had multiple potential glycosites. It is also pertinent to note that we performed tandem mass spectrometry only on single replicates and that too only for some of the mini-flagellin constructs and chimeras. We, therefore, relied on the intact mass analysis data for comparison of glycosylation among the different mini flagellin constructs and chimeras and on the tandem mass spectrometry data predominantly for the identification of glycosites. Another obfuscating factor was the presence of significant heterogeneity of glycosylation in *Cb*Fla co-expressed with *Cb*Maf. In addition to the previously reported seven Ser and five Thr glycosites[13,18], we identified two other potential glycosites (T197 and S199). Heterogeneity in glycosylation has previously been reported in

*Aeromonas caviae* flagellins, too[28,43]. The factors that affect heterogeneity of glycosylation are not known but the phenomenon hints at several Ser/Thr residues on the flagellin being similarly accessible and equally acceptable as substrates by the Maf glycosyltransferase. Sequence analysis of the amino acid neighborhood of *Cb*Fla around the glycosites identified in previous studies[13,18] and in our current study using the pLogo tool[44] hinted at the preponderance of small apolar or hydrophobic or polar uncharged residues (such as valine, methionine, isoleucine, phenylalanine, threonine, and glycine) immediately around the glycosite, flanked by polar charged or uncharged residues (such as arginine, lysine, aspartate, glutamate, asparagine, and glutamine) (Supplementary Fig. S3u–x); it is, therefore, likely that apolar residues are important for glycosylation by *Cb*Maf, similar to that observed in *A. caviae* flagellin[28].

E. coli, the popular cell factory for expressing recombinant proteins lacks endogenous glycosylation; outfitting *E. coli* with a glycosylation machinery is expected to facilitate the heterologous production of recombinant industrially useful or therapeutic glycoproteins in a low-cost, fast, and simple manner[45,46]. Bacterial protein glycosyltransferases with defined acceptor substrate features, such as the pilin-specific PglS and Tfpm oligosaccharyltransferases have been explored in glycoengineering applications and represent significant advances in the field[47–49]. Human mucin-derived peptides have also been O-glycosylated in *E. coli* cells heterologously co-expressing epimerase and GalNAc-T2 genes; ppGalNAc transferases glycosylate Ser/Thr residues in the vicinity of proline residues[50–52]. Bacterial oligosaccharyltransferases (OTases) such as the N-linked OTase PglB from *Campylobacter jejuni* and the O-linked OTase PglL from *Neisseria meningitides* recognize asparagine or serine residues within conserved sequons (D/E-X1-N-X2-S/T (where X1 and X2 are any amino acids except proline) and WPAAASAP, respectively) for glycosylation, and consequently such sequons might be easily incorporated into polypeptide sequences to outfit them for glycosylation[42,53–56]. Albeit the Maf glycosyltransferases do not recognize such clearly defined glycosylation sequons and consequently do not lend themselves to as facile applications, we have demonstrated here the robust glycosylation of mini-flagellin constructs and chimeras by *Cb*Maf in an *E. coli* lab strain that overexpresses CMP-sialic acid. We believe that our study paves an important first step towards the use of *Cb*Maf in exciting glycoengineering applications in the future. It is important to note, though, the KDO promiscuity, which might limit the implementation of this system where only Neu5Ac modifications are desired; *E. coli* strains that produce CMP-Neu5Ac without CMP-KDO or the use of engineered or other Maf glycosyltransferases that accept CMP-Neu5Ac but not CMP-KDO as donor substrate, will improve the usability of this system for applications where protein neo-O-sialylation (i.e., modifications with Neu5Ac with reasonable stringency) is sought. The ability of *Cb*Maf to identify and glycosylate the mini-flagellin constructs and chimeras paves the way for designing future experiments that can completely dissect the features of the acceptor substrate of *Cb*Maf, and thereby formulate approaches by which biotherapeutic protein neo-glycosylation by *Cb*Maf can become a reality. For instance, future experiments utilizing hybrid flagellin constructs (for instance, the *Cb*Fla secondary structure scaffold with the *Gk*FlaA2/*Gk*FlaA1 loop region, and vice versa) will be useful to dissect the stringency observed with respect to the cognate acceptor substrate. Other ideas for experiments planned for the future include a proteomics-based exploration of potential non-flagellin substrates of *Cb*Maf in *E. coli*, and structural alignment-based mining or custom design of non-flagellin proteins with loops of similar conformation as that observed locally around glycosylated Ser/Thr in *Cb*Fla and subsequent assessment of their suitability as *Cb*Maf acceptor substrates.

## Methods
### Bioinformatics analysis
The domain architectures of *Cb*Fla deletion constructs were represented using the Illustrator of Biological Sequences (IBS) software[26]. The structures of all the *Cb*Fla deletion protein constructs were predicted from locally installed Alphafold2 software[57]. The highest scoring structures, i.e., structural models with rank '0' were downloaded and visualized by Pymol

software, Molecular Graphics system, version 2.0 (Schrödinger, LLC). AlphaFold2 predicted structures of all proteins used in this work were also visualized with reverse rainbow coloring as per pLDDT scores in PyMOL (Schrodinger) Supplementary Fig. S7 and the structural models are provided in Supplementary Data 2. Glycosite analysis was conducted using the online pLogo tool[44] with manually curated 11-amino acid peptide sequences containing glycosylated or unglycosylated Ser/Thr, and a set of all 11-amino acid peptide sequences of CbFla generated using a Scratch (MIT)[58] code kindly developed and provided by Ms. Ipshita Srikrishna, Chandigarh.

### Cloning of recombinant mini-flagellin constructs and chimeras and *Cb*Maf
The expression vector pET-28a(+) was used for cloning *Cb*Fla deletion constructs and chimeras with vector-encoded C-terminal hexahistidine tags. The respective primers used for PCR amplification of each gene are listed in Supplementary Table S2. The NcoI- and XhoI- restriction digested PCR amplicons were ligated with NcoI- and XhoI- restriction digested and calf intestinal phosphatase (CIP)-treated pET-28a(+) DNA using T4 DNA Quick ligase at room temperature for 7 minutes. The ligation mix was transformed in TOP10 competent cells, and subsequently, the transformed colonies were screened for positive clones by Sanger sequencing using T7 oligonucleotide primers (In-house DNA Sequencing facility, CSIR-IMTECH).

The construct D0A7loop was cloned in pET-28a(+) vector using NEBuilder HiFi DNA assembly. In brief, three gene fragments with overlapping ends were PCR amplified and purified by excision and extraction following agarose gel electrophoresis. The pET-28a(+) vector was linearized by digestion with NcoI and XhoI restriction enzymes. The gene fragments and linearized vector were then added to 10 μL of Gibson Assembly master mix and incubated at 50 °C for 1 h. After incubation, 2 μL of the mix was transformed in TOP10 competent cells. The transformed colonies were screened for positive clones by Sanger DNA sequencing using T7 oligonucleotide primers (In-house DNA Sequencing facility, CSIR-IMTECH).

### *Cb*Maf was cloned and expressed from the expression vector, pCDF-Duet-1
All expression constructs used in this work are being deposited in the plasmid repository of Microbial Type Culture Collection and Gene Bank (MTCC), CSIR- Institute of Microbial Technology, Chandigarh, India.

### Expression and purification of *Cb*Fla deletion constructs and *Cb*Fla chimeras in *E. coli* EV136 cells
We received *E. coli* EV136 cells as a kind gift from Prof. Eric R. Vimr. The *Cb*Fla deletion constructs and *Cb*Fla chimeras cloned in pET-28a(+) wered transformed in *E. coli* EV136 cells for protein expression and purification. For co-expression, *Cb*Fla constructs in pET-28a(+) and *Cb*Maf in pCDFDuet-1 were co-transformed in *E. coli* EV136 strain. Since both the *Cb*Fla deletion constructs and *Cb*Maf were cloned in T7 expression vectors, the Targe Tron pAR1219 vector (Sigma) that encodes T7 RNA polymerase under lac UV5 promoter was used for expression in *E. coli* EV136 cells. The culture medium used was minimal medium with M63 salts first made with potassium dihydrogen phosphate, ammonium sulfate and ferrous sulfate, with pH adjusted to 7.0 and autoclaved. The other supplements of the medium - 1 mM MgCl2, 0.2% glycerol, and casamino acids - were added at the time of bacterial inoculation in the medium for primary and secondary cultures. The medium also contained the antibiotics, kanamycin, spectinomycin, and ampicillin for selection.

A single colony was picked and inoculated in minimal medium and grown overnight at 37 °C with 200 rpm to obtain the primary culture. Using 1% of primary culture as inoculum, secondary cultures were grown at 37 °C with 200 rpm shaking until the cell density was equivalent to $OD_{600}$ of 0.6–0.8, whereupon the recombinant protein expression was induced with 0.5 mM isopropyl β-D-1-thiogalactoside (IPTG) and the culture was incubated at 22 °C with 180 rpm shaking for 10 h.

The hexahistidine-tagged recombinant *Cb*Fla constructs were purified by Ni-NTA metal ion affinity chromatography. For purification, the cell pellet from 200 mL culture was re-suspended in 10 mL Tris buffered saline (20 mM Tris, pH 7.5 and 150 mM NaCl) to make a homogeneous cell suspension. N-laurylsarkosine was added at a final concentration of 0.5% to achieve complete lysis. After re-suspension, the cells were lysed by sonication using an ultrasonicator (Sonics & Materials INC) for 10 min (pulse 10 s on and 10 s off, amplitude 30%). The cell lysate was then centrifuged at 11,000 rpm for 30 min. The clarified lysate obtained was incubated with Ni-NTA metal ion affinity His-bind resin at 4 °C for 1 hour on a rotary mixer (Rotospin) with end-over-end rotation.

For partial purification of *Cb*Fla, *Cb*FlaA1, *Cb*FlaA5, *Cb*FlaA7 and *Cb*FlaA11, each singly expressed or co-expressed with *Cb*Maf, the Ni-NTA column was washed with 40 mM imidazole in TBS and eluted with 250 mM imidazole. On the other hand, *Cb*FlaA3, *Cb*FlaA4, *Cb*FlaA4_A5, *Cb*FlaA3_A7, *Cb*FlaA4_A7, *Cb*FlaA7v, *Cb*FlaD0A7loop, SrNaFLD, *Cb*FlaA7vSrNaFLD, *Sr*Na*Cb*FlaA1FLD, *Cb*FlaA1*Sr*NaFLD, GKGAKS_*Sr*NaFLD and *Cb*Fla_A1helix_GKGAKS_*Sr*NaFLD, each singly expressed or co-expressed with *Cb*Maf, were enriched using step gradient elution of 20 mM, 50 mM, 100 mM, 250 mM and 500 mM imidazole in TBS.

We used SDS-PAGE followed by Coomassie blue staining to assess protein purity.

## Expression and purification of flagellins with non-cognate Maf glycosyltransferases in *E. coli* EV136 cells

*Cb*Fla (in pET28a(+) vector) was co-expressed with *Gk*Maf (in pCDFDuet-1 vector) in *E. coli* EV136 strain. *Gk*FlaA1 and *Gk*FlaA2 (in pET28a(+)) vectors were also co-expressed with *Cb*Maf (in pCDFDuet-1 vector) in *E. coli* EV136 strain. The Targe Tron pAR1219 vector encoding T7 RNA polymerase was used for expression of flagellins and Mafs in *E. coli* EV136 cells. The culture medium used for the expression was minimal medium with similar salts as described for the co-expression of *Cb*Fla constructs with *Cb*Maf. Kanamycin, spectinomycin, and ampicillin antibiotics were added to the medium for selection. All the expression conditions were same as described for the co-expression of *Cb*Fla constructs with *Cb*Maf.

The hexahistidine-tagged recombinant *Cb*Fla (co-expressed with *Gk*Maf), *Gk*FlaA1 and *Gk*FlaA2 (each co-expressed with *Cb*Maf) were purified by Ni-NTA metal ion affinity chromatography. For purification, the cell pellet from 200 mL culture was re-suspended in 10 mL Tris buffered saline (20 mM Tris, pH 7.5 and 150 mM NaCl). N-laurylsarkosine was added at a final concentration of 0.5% to achieve complete lysis. After re-suspension, the cells were lysed by sonication using an ultrasonicator (Sonics & Materials INC) for 10 minutes (pulse 10 s on and 10 s off, amplitude 30%). The cell lysate was then centrifuged at 11,000 rpm for 30 min. The clarified lysate obtained was incubated with Ni-NTA metal ion affinity His-bind resin at 4 °C for 1 h on a rotary mixer (Rotospin) with end-over-end rotation.

For partial purification of *Cb*Fla (co-expressed with *Gk*Maf), *Gk*FlaA1 and *Gk*FlaA2 (each co-expressed with *Cb*Maf), the Ni-NTA column was washed with 40 mM imidazole in TBS and eluted with 250 mM imidazole. We used SDS-PAGE followed by Coomassie blue staining to assess protein purity.

## On-blot PAL and Western analysis

We performed Western blotting of *Cb*Fla deletion constructs and chimeras and on-blot biotinylation using the PAL procedure described by Zeng et al.[32] to detect the presence of protein modification by sialic acid[32]. The hexahistidine-tagged *Cb*Fla mini flagellin constructs and chimeras were detected by mouse anti-His tag monoclonal antibody, and *Cb*Maf was detected using mouse anti-S-tag monoclonal antibody. Both the antibodies were used at a dilution of 1:5000. HRP-conjugated donkey anti-mouse IgG was used as the secondary antibody at a dilution of 1:10000. For PAL labeling, the sialic acid residues were subjected to oxidation by 1 mM sodium metaperiodate (NaIO4) in phosphate buffered saline (PBS; 20 mM phosphate buffer, pH 7.4 with 150 mM NaCl) at 4 °C for 20 min. This was

followed by a quenching step using 2 mM glycerol, washes with PBS, and incubation with PBS, pH 6.7, containing 250 µM aminooxy-biotin (Invitrogen, Carlsbad, California) and 10 mM aniline at room temperature in the dark for 60 min. The blot was then blocked overnight in 5% skimmed milk and probed with HRP-conjugated streptavidin and chemiluminescence was detected using Luminata Forte western HRP substrate. The blot was visualized in Image Quant LAS4000 (GE Healthcare Life Sciences) and G:BOX Chemi XRQ gel doc system (Discovery Scientific Solutions). Some blots were physically trimmed following transfer to remove very high or low molecular mass regions (after considering the molecular mass of the proteins of interest) to save on reagents for biotinylation, and the molecular mass range of proteins present in the blot is indicated by the molecular mass marker on the blot. Digitally cropped regions of all blots used in this work are indicated in Supplementary Fig. S8.

## Acetone precipitation

Purified flagellin constructs and chimeras were acetone precipitated for mass spectrometry analysis. The imidazole eluates of *Cb*Fla constructs obtained following Ni-NTA metal ion affinity chromatography were added to pre-chilled acetone at 1:4 ratio and kept at −80 °C overnight. The samples were then thawed, centrifuged at 12,000 rpm for 10 minutes at 4 °C and the protein precipitate washed thrice with 80% acetone and stored at −80 °C until analysis by mass spectrometry.

## Mass spectrometry analysis for intact mass measurement

Acetone precipitated protein pellets retrieved from storage at −80 °C were washed four times with 80% acetone (in water), dissolved in 0.1% TFA, and subjected to intact mass analysis using Electrospray Ionization (ESI) - Liquid chromatography (LC)/Mass Spectrometry (MS) on an Agilent G6550A Quadrupole Time of Flight (Q-TOF) mass spectrometer (In-house mass spectrometry facility). For LC, a Zorbax 300SB-C18 column (Agilent Technologies, Santa Clara, California) was used at a flow rate of 0.4 mL/min with acetonitrile and water as mobile phase solvents in 0.1% formic acid (solvent A-water, solvent B-acetonitrile). Sample components were separated in solvent B gradient (30–95%) by LC and infused at a scan rate of 1 scan/min in the MS Q-TOF mass spectrometer for 23 minutes with acquisition range 400–3200 m/z, and ionization source dual Agilent Jet Spray electrospray ionization (ESI) in positive ion mode. The data was analyzed using Agilent Mass-Hunter software. *Cb*Fla, and the *Cb*Fla mini-flagellin constructs and chimeras eluted between 7.4 and 9 min. The abundance of unmodified and various modified protein species in the spectra was determined by measuring the peak intensity heights. Raw data for all spectra plotted in this work are available in Supplementary Data 3.

## Tandem mass spectrometry for glycosite identification

*Cb*Fla, the mini-flagellin constructs (*Cb*Fla, *Cb*FlaA1, *Cb*FlaA3, *Cb*FlaA4, *Cb*FlaA5, *Cb*FlaA7, *Cb*FlaA4_A5 and *Cb*FlaD0A7loop) and the chimeras (*Sr*NaFLD, *Cb*FlaA1_*Sr*NaFLD, GKGAKS-*Sr*NaFLD and *Cb*Fla_A1helix-GKGAKS_*Sr*NaFLD), co-expressed with *Cb*Maf, were also subjected to tryptic digestion followed by tandem mass spectrometry (performed at Taplin Biomedical Mass Spectrometry Facility, Harvard Medical School, USA). For sample preparation, we resolved the samples on SDS-PAGE and further stained them using Coomassie Blue. Gel bands of the proteins of interest were then excised and subjected to in-gel protease digestion with trypsin, Briefly, the gel pieces were dehydrated with acetonitrile (10 min), rehydrated with 50 mM ammonium bicarbonate, reduced with 1 mM DTT in 50 mM bicarbonate buffer (30 min at 60 °C), alkylated with 5 mM iodoacetamide in 50 mM bicarbonate buffer (15 mins at room temperature in the dark), the reaction quenched with 5 mM DTT, and sequence grade trypsin (5 ng/µL) added (overnight incubation at 37 °C). The trypsin-digested peptides were extracted, desalted and reconstituted in 5-10 µL of solvent A (97.5% water and 2.5% acetonitrile in 0.1% formic acid). A nano-scale reverse-phase HPLC capillary column made by filling a fused 30 cm long silica capillary of inner diameter 100 µm with 2.6 µm C18 spherical silica beads[59], was equilibrated and the tryptic digest sample loaded onto the

column. Peptides were eluted with increasing amounts of solvent B (97.5% acetonitrile and 2.5% water, in 0.1% formic acid), and subjected to electrospray ionization in a Linear Trap Quadrupol (LTQ) Orbitrap Velos Pro ion-trap mass spectrometer (Thermo Fisher Scientific, Waltham, Massachusetts). Tandem mass spectra were obtained by identifying, extracting and fragmenting distinct peptides.

The software programme SEQUEST (Thermo Fisher Scientific, Waltham, Massachusetts) was used to compare the protein database with the acquired fragmentation pattern in order to determine peptide sequences (and subsequently, protein identity)[60]. All the query databases of the samples also had a reverse version of their sequences, and data was filtered such that the peptide False Detection Rate (FDR) was under 2%. The variable methionine modification of 15.994915 Da was allowed by the search parameters. The Ser/Thr residues were examined for alterations of 291 Da (Neu5Ac) or 220 Da (KDO) using the Modscore algorithm. The fragment ion tolerance parameters were 0.6 Da for Site 1 Position A and 0.3 Da for Site 1 Position B, respectively. If the Site 1 Position A and Site 1 Position B Modscore values for the same site were both >19, then the location was regarded as confidently assigned. If a peptide had a single Ser/Thr residue, the Modscore value was 1000 for confident assignment of site modification[61]. Residue numbering and glycosites of *Cb*Fla, and the miniflagellin constructs and chimeras are as per the amino acid sequence of the recombinant proteins, which contain an additional alanine residue following the N-terminal methionine residue and a C-terminal hexahistidine tag.

## Statistics and reproducibility
Three biological replicates (independent protein preparations) were used for intact mass analyses, and the results were reproducible. Only one biological replicate was used for the tandem mass spectrometry. No statistical tests were used in the study.

## Reporting summary
Further information on research design is available in the Nature Portfolio Reporting Summary linked to this article.

## Data availability
The mass spectrometry proteomics data reported in the manuscript have been deposited to the ProteomeXchange Consortium[62] via the PRIDE[63] partner repository with the dataset identifier PXD044191. Data supporting the findings in this article are available within the article and supplementary information, and available from the corresponding author upon reasonable request. All expression constructs used in this work are being deposited in the plasmid repository of Microbial Type Culture Collection and Gene Bank (MTCC), CSIR- Institute of Microbial Technology, Chandigarh, India.

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

## Acknowledgements

The authors thank Prof. Eric R. Vimr, University of Illinois at Urbana-Champaign for the *E. coli* strains, EV136, EV36, and EV240. The authors acknowledge the Mass Spectrometry facility at CSIR-Institute of Microbial Technology, Chandigarh, and the Taplin Biological Mass Spectrometry facility at Harvard Medical School for their mass spectrometry services, Ms. Madhu Lata and Ms. Anu Walia, CSIR-Institute of Microbial Technology, Chandigarh, for technical assistance in data compilation and plasmid archival, respectively, and Ms. Ipshita Srikrishna, Chandigarh, for kindly developing and providing a Scratch (MIT) code for generating 11-amino acid peptide sequences of *Cb*Fla for glycosite analysis by pLogo. This work was supported by the Science and Engineering Research Board, Department of Science and Technology, Government of India (grant no. EMR/2016/006866 to RTNC and Sr.S.). A.K. and So. S. acknowledge UGC for their fellowships. The authors acknowledge CSIR-IMTECH (manuscript communication number: 010/2024) for the research facilities and infrastructure.

## Author contributions

So.S.: Methodology, Investigation (most of the biochemical experiments), Validation, Formal analysis, Writing – Original Draft, Writing – Review & Editing, A.K.: Methodology, Investigation (a few biochemical experiments), Formal analysis, Writing – Review & Editing, Sr.S.: Methodology, Software, Investigation (protein structure predictions using AlphaFold2), Formal analysis, Writing – Review & Editing, Funding Acquisition TNCR: Conceptualization, Methodology, Formal analysis, Writing – Original Draft, Writing – Review & Editing, Supervision, Project administration, Funding acquisition.

## Competing interests

The authors declare no competing interests.
