## [Peer Review File · Communications Biology]

Referee expertise:

Referee #1: expert in O-glycosylation of flagellin

Referee #2: expert in Chemical glycobiology

Referee #3: expert in Chemical glycobiology

Reviewers' comments:

Reviewer #1 (Remarks to the Author):

The work described here may bear some relevance from a protein engineering perspective, but it is otherwise not physiologically interesting or important.

It reports the construction of mini substrate for glycosylation by the Maf flagellin glycosyltransferase, but with substrate that is artificial. It is not even clear what the true substrate is, presumably legionaminic acid, so whatever this study tells us about recognition of the acceptor only holds true for the non-natural ones, KDO and sialic acid.

In principle it is no surprise that the determinants surrounding the D1 loop, i.e. the D1 domain are the most important areas in the substrate for the Maf to bind its substrate. Trimming from both ends reveals that the extremities i.e. the D0 is not required. The fact that a reduced loop can be made is interesting, however we still know little about how the Maf recognize the substrates and how it glycosylates.

The big achievement here is that instead of full-length protein it is now possible to graft a half-flagellin sequence to a protein of interest for glycosylation, but the glycosylation remains on the flagellin part. Finally, the fact that other flagellins are not glycosylated by Maf is neither surprising nor conclusive. Are these flagellins bound by Maf these Mafs or does it not work because it would only work in this configuration if the correct (natural) donor sugar is available? It is known from other systems that flagellins are normally not glycosylated by heterologous expression of Maf from different genus or vice versa.

The absence of line numbers makes it hard to refer to spelling mistakes at specific positions and the paper is not an easy read.

comments

“... Maf glycosyltransferase from *Magnetospirillum magneticum* comprises three domains, of which the central Maf_flag10 domain ... “ – what do the other two domains do?

“ ... Maf glycosyltransferases display donor substrate promiscuity ...” – any enzyme will reveal some sort of promiscuity, especially in the absence of the preferred substrate. the question is how the promiscuity compares in the presence of the natural donor. If the enzyme chooses the wrong substrate when the normal donor is around, then once can talk about promiscuity, but if the enzyme only uses the alternative donor when there is no or little natural donor, then I wonder is this is relevant. With other words the promiscuity should be quantified by enzyme specificity and not referred

Reviewer #2 (Remarks to the Author):

In this manuscript, the authors report an extensive study of the structural attributes of CbFlaA that are required for glycosylation by CbMaf in a heterologous host (*E.coli*). This work builds on a previous study published by the authors, showing that CbMaf can glycosylate serine and threonine residues in CbFlaA with Neu5Ac and KDO, when co-expressed in *E.coli*. The authors prepare a series of CbFlaA constructs which contain various lengths of the CbFlaA D1 loop region and/or the D0/D1 helices. These are co-expressed both with and without CbMaf, in an *E.coli* strain that accumulates CMP-Neu5Ac. Glycosylation of CbFlaA proteins (with Neu5Ac and KDO) following their purification, is then assessed by using a combination of experiments including on-blot PAL, intact protein mass spectrometry and MS/MS following protease digestion. The authors show that CbMaf can glycosylate mini flagellin constructs containing shortened alpha-helical structural scaffolds and reduced surface-accessible loop regions. It was shown that CbMaf recognition & glycosylation of surface accessible Ser/Thr residues was dependant on their correct positioning by the supporting secondary structure scaffold. The authors also show that CbMaf can glycosylate flagellin chimeras, made by fusing truncated flagellin sequences to an unrelated beta-sandwich protein. Overall the manuscript is well written and includes a significant body of work which supports the authors findings. It should be published in *Communications Biology*, with some minor corrections detailed below.

- Figure 1d. – Is the middle panel validating the expression of CbMaF (i.e. probing for the S-tag)? This should be clarified in the figure legend, as middle panel is not described.

- Supplementary data 2, Figure 1 – “The image in the outlined boxes from each blot was used in Figure 1c” –should this be Figure 1d?
- Supplementary data 2, Figure 1 – left hand panel (anti-His tag blot) – what do lanes 5 & 6 show (multiple anti-His reactive bands) as these are not described in the figure legend.
- Figure 1. e & f. – it would be useful to include larger images of the raw (before charge deconvolution) protein MS data (currently included in the figure, but too small to read) in the supplementary information. The same is true for the other protein MS spectra reported in the manuscript.
- Supplementary Figure S4a2- MS/MS analysis of glycosylated CbFlaA4, on residues where both modifications are present, can you make sure the symbol is above the modified residue. There is a # above a glycine in this sequence. Correct throughout Figure S4.
- For the assignment of glycosylation sites within the CbFlaA proteins after trypsin/GluC digest throughout the manuscript, assigned MS/MS spectra should be included in the supporting information (at least one high confidence spectrum validating each glycosite). Indicate details of the peptide sequence, modified site & modification (Neu5Ac/KDO) to that site.
- Fig 3. c – the On-blot PAL showing low-level modification of CbFlaA5 is very difficult to see, can the contrast on this image be adjusted to have a lighter background?
- Additional supplementary Data 2 Fig. 3m On-blot PAL of CbFlaA11 – this western blot looks a little strange to me – there is a strange white imprint where you might expect to see labelled CbFlaA11. Can you be sure that there is no modification detected here?
- For the western blots reported in Additional supplementary Data 2 where two panels are shown for a particular western (those related to Fig. 1c, 2c, 2h, 3c, 3h, 3m, 4c, 4h, 4m, 4r, 5c, 5h, 5m, etc...), could you please clarify what each panel is showing. Is this just inversion of the contrast?
- You report that the T124 residue in singly expressed CbFlaA5 was identified as glycosylated – did manual inspection & assignment of the MS/MS spectrum validate this observation or is this a false positive?
- Could you clarify why you performed a Glu-C digest & MS/MS analysis of CbFlaA5 & CbFlaA7. Also, why were significantly fewer glycosylation sites assigned after using this protease? Please include a statement in the main text explaining why the Glu-C digest & MS/MS analysis was performed, in addition to the tryptic digests for these proteins.
- In the discussion, the statement “The remarkable acceptor substrate promiscuity...” – is this really acceptor substrate promiscuity? Firstly, CbMaf won’t glycosylate GfFlaA. What you have done here is define the minimal recognition determinants of CbFlaA to be glycosylated. by CbMaf. Please rephrase in discussion.

Reviewer #3 (Remarks to the Author):

In this paper, Sunsunwal et al. explore the sugar acceptor protein substrate scope for a *Clostridium botulinum* flagellin O-glycosyltransferase using a heterologous *E. coli* expression system. The authors constructed a series of truncated flagellin mutants to assess how different regions of flagellin contribute to O-glycosylation. They found that the ND1b-D1 region is minimally required for glycosylation, while the ND1a2 and CD1_2 regions enhance its efficacy. Notably, the study observed compensatory effects where shortening the D1 loop, which structurally masks the ND1 and CD1 regions, led to glycosylation at non-canonical Ser/Thr residues in these regions. This finding supports the importance of surface accessibility for glycosylation sites. Technically, the authors used engineered *E. coli* EV136 cells for effective O-glycosylation with Neu5Ac and KDO, similar to their previous work (*Glycobiology*, 31(3), 288-306, 2021). This approach demonstrates potential for producing artificial glycoproteins containing beta-sandwich domains. Additionally, the authors employed a PAL method for detecting *in vivo* glycosylation, developed by Y. Zeng, T. N. C. Ramya, and colleagues (*Nature Methods*, 6, 207-209, 2009). Intact mass spectrometry proved useful in detecting glycosylations that the PAL method could not identify.

Overall, the experimental data provide invaluable insights into the sequence/structure attributes of the glycosylated substrate, which are crucial for the field of future glycoprotein engineering. The authors conducted comprehensive structure predictions using AlphaFold, which offered structural insights into the glycosylated substrate. However, the manuscript lacks statistical analyses of the amino acid sequences surrounding glycosylation sites, despite experimental identification of these residues by tandem mass spectrometry.

Questions and Suggestions:

Major Point #1: Supplementary Figure 4: To clearly validate the identification of glycosylation sites for readers, it would be beneficial to include MS/MS spectra with peak assignments for all glycosylation sites, especially for CbFla and CbFlaA5. Consider using formats similar to those presented in Figures 7C and 7D of the authors' previous study (*Glycobiology*, 31(3), 288-306, 2021).

Major Point #2: On page 3, the second paragraph of the introduction mentions that "...the *Aeromonas caviae* Maf glycosyltransferase modifies Ser/Thr residues flanked by hydrophobic amino acids such as leucine and isoleucine (28)." For CbMaf, is there a preference for any specific amino acid sequence around the glycosylated residues? If so, is there a correlation with physicochemical properties such as van der Waals volume, hydrophobicity, and electrical charge? Employing a tool like pLogo for statistical analysis of

the glycosylation sites could be particularly beneficial (refer to Nature Methods, 10, 1211-1212, 2013).

Minor Points:

Minor Point #3: On page 3, the second paragraph of the introduction states, "...the *A. caviae* Maf glycosyltransferase interacts with both unglycosylated and glycosylated flagellin (29)." Please verify the accuracy of this citation.

Minor Point #4: It would be helpful if Figures 1b, 2b, 2g, 2l, 3b, 3g, 3l, 4a, 4f, 4k, 4p, 5a, 5f, and 5k clearly indicated the identified glycosylation sites on the protein region diagrams.

Minor Point #5: In Figures 1f, 2e, 2j, 2o, 3e, 3j, 3o, 4e, 4j, 4o, 4t, 5e, 5j, and 5o, substituting the labels "N" and "K" with "Neu5Ac" and "KDO," respectively, would clarify and prevent confusion with the potassium adduct.

Point-by-point Responses to Referees' comments

We are very thankful to all three reviewers for the rigorous and thorough review of our manuscript and for the suggestions, which we feel have helped improve our manuscript. Our point-by-point responses to the reviewers' comments follow below.

Reviewer #1 (Expert in O-glycosylation of flagellin) (Remarks to the Author):

The work described here may bear some relevance from a protein engineering perspective, but it is otherwise not physiologically interesting or important. It reports the construction of mini substrate for glycosylation by the Maf flagellin glycosyltransferase, but with substrate that is artificial. It is not even clear what the true substrate is, presumably legionaminic acid, so whatever this study tells us about recognition of the acceptor only holds true for the non-natural ones, KDO an sialic acid. In principle it is no surprise that the determinants surrounding the D1 loop, i.e. the D1 domain are the most important areas in the substrate for the Maf to bind its substrate. Trimming from both ends reveals that the extremities i.e. the D0 is not required. The fact that a reduced loop can be made is interesting, however we still know little about how the Maf recognize the substrates and how it glycosylates. The big achievement here is that instead of full-length protein it is now possible to graft a half-flagellin sequence to a protein of interest for glycosylation, but the glycosylation remains on the flagellin part. Finally, the fact that other flagellins are not glycosylated by Maf is neither surprising nor conclusive. Are these flagellins bound by Maf these Mafs or does it not work because it would only work in this configuration if the correct (natural) donor sugar is available? It is known from other systems that flagellins are normally not glycosylated by heterologous expression of Maf from different genus or vice versa.

The absence of line numbers makes it hard to refer to spelling mistakes at specific positions and the paper is not an easy read.

Thank you for the comments and suggestions. We have inserted line numbers and checked for spelling mistakes.

Comments

"... Maf glycosyltransferase from Magnetospirillum magneticum comprises three domains, of which the central Maf_flag10 domain ... " – what do the other two domains do?

The putative roles of the other two domains have now been added to the introduction section of the manuscript (lines 62-68, pg. 3). The N-terminal domain adopts a degenerated Rossmann-like fold with five alpha helices flanking parallel beta-sheets of four beta strands, and the C-terminal domain is composed mainly of alpha-helical bundles. The function of these domains remains unclear, although the N-terminal domain shows weak similarities with some motifs in methyltransferases and the C-terminal domain shows weak similarities with flagellin export chaperones (such as FliS in Aquifex aeolicus) and with flagellins (such as FliC in Burkholderia pseudomallei) (Sulzenbacher et al, 2018).

“ ... Maf glycosyltransferases display donor substrate promiscuity ...” – any enzyme will reveal some sort of promiscuity, especially in the absence of the preferred substrate. the question is how the promiscuity compares in the presence of the natural donor. If the enzyme chooses the wrong substrate when the normal donor is around, then once can talk about promiscuity, but if the enzyme only uses the alternative donor when there is no or little natural donor, then I wonder is this is relevant. With other words, the promiscuity should be quantified by enzyme specificity and not referred.

Thank you for raising this interesting point. We agree that it would add value to be able to compare the donor substrate promiscuity in the presence of the natural donor substrate, in this case, CMP-legionaminic acid for *Clostridium botulinum* Maf. However, this is beyond the scope of this study since the *E. coli* EV136 strain we use does not produce CMP-legionaminic acid and this study focuses on the acceptor substrate rather than the donor substrate. Further, we would like to emphasize that having the enzyme choose the alternative donor substrate (CMP-sialic acid) when there is no natural donor substrate (CMP-legionaminic acid) is relevant for glycoengineering applications, where it would be desirable to decorate recombinant therapeutics with sialic acid (a monosaccharide abundant in humans) but not with legionaminic acid (not found in humans).

Reviewer #2 (Expert in Chemical glycobiology) (Remarks to the Author):

In this manuscript, the authors report an extensive study of the structural attributes of CbFlaA that are required for glycosylation by CbMaf in a heterologous host (E.coli). This work builds on a previous study published by the authors, showing that CbMaf can glycosylate serine and threonine residues in CbFlaA with Neu5Ac and KDO, when co-expressed in E.coli. The authors prepare a series of CbFlaA constructs which contain various lengths of the CbFlaA D1 loop region and/or the D0/D1 helices. These are co-expressed both with and without CbMaf, in an E.coli strain that accumulates CMP-Neu5Ac. Glycosylation of CbFlaA proteins (with Neu5Ac and KDO) following their purification, is then assessed by using a combination of experiments including on-blot PAL, intact protein mass spectrometry and MS/MS following protease digestion. The authors show that CbMaf can glycosylate mini flagellin constructs containing shortened alpha-helical structural scaffolds and reduced surface-accessible loop regions. It was shown that CbMaf recognition & glycosylation of surface accessible Ser/Thr residues was dependant on their correct positioning by the supporting secondary structure scaffold. The authors also show that CbMaf can glycosylate flagellin chimeras, made by fusing truncated flagellin sequences to an unrelated beta-sandwich protein. Overall the manuscript is well written and includes a significant body of work which supports the authors findings. It should be published in Communications Biology, with some minor corrections detailed below.

Thank you for the comments and suggestions.

• Figure 1d. – Is the middle panel validating the expression of CbMaF (i.e. probing for the S-tag)? This should be clarified in the figure legend, as middle panel is not described.

Thank you for pointing this out. Yes, the middle panel probes the expression of *CbMaf* as detected with an anti-S tag antibody. This has been added to the legend in the revised manuscript (lines 959-960, pg. 26).

- *Supplementary data 2, Figure 1 – “The image in the outlined boxes from each blot was used in Figure 1c” –should this be Figure 1d?*

Thank you for pointing this out. Yes, it should be Figure 1d, and it has been changed in the revised **Additional Supplementary Data 2**, Figure 1.

- *Supplementary data 2, Figure 1 – left hand panel (anti-His tag blot) – what do lanes 5 & 6 show (multiple anti-His reactive bands) as these are not described in the figure legend.*

The data for lanes 5 and 6 in the uncropped blot do not correspond to the data for this manuscript and are, therefore, not mentioned. Kindly ignore lanes 5 and 6.

- *Figure 1. e & f. – it would be useful to include larger images of the raw (before charge deconvolution) protein MS data (currently included in the figure, but too small to read) in the supplementary information. The same is true for the other protein MS spectra reported in the manuscript.*

We have increased the size of the inset images as much as possible and also increased the font size of the labels in all the deconvoluted spectra (**Figures 1-5**).

- *Supplementary Figure S4a2- MS/MS analysis of glycosylated CbFlaA4, on residues where both modifications are present, can you make sure the symbol is above the modified residue. There is a # above a glycine in this sequence. Correct throughout Figure S4.*

Thank you for pointing this out. We have corrected **Supplementary Figure S4** and adjusted the symbols over Serine/Threonine.

- *For the assignment of glycosylation sites within the CbFlaA proteins after trypsin/GluC digest throughout the manuscript, assigned MS/MS spectra should be included in the supporting information (at least one high confidence spectrum validating each glycosite). Indicate details of the peptide sequence, modified site & modification (Neu5Ac/KDO) to that site.*

A representative MS/MS spectrum has been added for each glycosite in the revised **Supplementary Figure S4**.

- *Fig 3. c – the On-blot PAL showing low-level modification of CbFlaA5 is very difficult to see, can the contrast on this image be adjusted to have a lighter background?*

We have adjusted the image in **Figure 3c** to have a lighter background and higher contrast in the revised manuscript.

- *Additional supplementary Data 2 Fig. 3m On-blot PAL of CbFlaA11 – this western blot looks a little strange to me – there is a strange white imprint where you might expect to see labelled CbFlaA11. Can you be sure that there is no modification detected here?*

This is likely an artifact. There is a clear signal in another lane (although not corresponding to data included here) indicating that the western has worked. Further, the white bands are also visible corresponding to the molecular marker bands and in the lanes where CbFlaA11 is not co-expressed with CbMaf (where modification is not expected), so we are fairly certain there is no modification here.

- For the western blots reported in Additional supplementary Data 2 where two panels are shown for a particular western (those related to Fig. 1c, 2c, 2h, 3c, 3h, 3m, 4c, 4h, 4m, 4r, 5c, 5h, 5m, etc...), could you please clarify what each panel is showing. Is this just inversion of the contrast?

The lower/right panels are the composite images (visible light plus chemiluminescence) showing the pre-stained molecular marker along with the inverted images of the upper/left panels. We have clearly indicated this for each figure in the revised **Additional Supplementary Data 2**.

- You report that the T124 residue in singly expressed CbFlaA5 was identified as glycosylated – did manual inspection & assignment of the MS/MS spectrum validate this observation or is this a false positive?

Manual inspection of the latter did not indicate the presence of glycan-modified peptide ions confirming glycosylation in singly expressed CbFlaA5. So, it could be a false positive. We have indicated this in the revised manuscript, **lines 230-232, pg 8**.

- Could you clarify why you performed a Glu-C digest & MS/MS analysis of CbFlaA5 & CbFlaA7. Also, why were significantly fewer glycosylation sites assigned after using this protease? Please include a statement in the main text explaining why the Glu-C digest & MS/MS analysis was performed, in addition to the tryptic digests for these proteins.

The tandem mass spectrometry for CbFlaA5 and CbFlaA7 was done prior to the other constructs. We tried two different proteases for digestion to assess the one with better coverage, site assignment, and spectra results. Since we obtained better results with trypsin digestion, we proceeded further with only trypsin digests for the remaining constructs. This has been included in **lines 243-246, pg 8** of the revised manuscript.

- In the discussion, the statement “The remarkable acceptor substrate promiscuity...” – is this really acceptor substrate promiscuity? Firstly, CbMaf won’t glycosylate GfFlaA. What you have done here is define the minimal recognition determinants of CbFlaA to be glycosylated by CbMaf. Please rephrase in discussion.

Thank you for pointing this out. We agree that what we have done is to define the minimal recognition determinants of CbFlaA to be glycosylated by CbMaf, and we have accordingly rephrased the statement in the revised manuscript, **lines 535-536, pg. 16**.

Reviewer #3 (Expert in Chemical glycobiology) (Remarks to the Author):

In this paper, Sunsunwal et al. explore the sugar acceptor protein substrate scope for a Clostridium botulinum flagellin O-glycosyltransferase using a heterologous E. coli expression system. The authors constructed a series of truncated flagellin mutants to assess how different regions of flagellin contribute to O-glycosylation. They found that the ND1b-D1 region is minimally required for glycosylation, while the ND1a2 and CD1_2 regions enhance its efficacy. Notably, the study observed compensatory effects where shortening the D1 loop, which structurally masks the ND1 and CD1 regions, led to glycosylation at non-canonical Ser/Thr residues in these regions. This finding supports the importance of surface accessibility for glycosylation sites. Technically, the authors used engineered E. coli EV136 cells for effective O-glycosylation with Neu5Ac and KDO, similar to their previous work (Glycobiology, 31(3), 288-306, 2021). This approach demonstrates potential for producing artificial glycoproteins containing beta-sandwich domains. Additionally, the authors employed a PAL method for detecting in vivo glycosylation, developed by Y. Zeng, T. N. C. Ramya, and colleagues (Nature Methods, 6, 207-209, 2009). Intact mass spectrometry proved useful in detecting glycosylations that the PAL method could not identify.

Overall, the experimental data provide invaluable insights into the sequence/structure attributes of the glycosylated substrate, which are crucial for the field of future glycoprotein engineering. The authors conducted comprehensive structure predictions using AlphaFold, which offered structural insights into the glycosylated substrate. However, the manuscript lacks statistical analyses of the amino acid sequences surrounding glycosylation sites, despite experimental identification of these residues by tandem mass spectrometry.

Thank you for the comments and suggestions. We have now used the pLogo tool to analyze the glycosites of full-length CbFla, using the tandem mass spectrometry data available from this and earlier studies (see below).

Questions and Suggestions:

Major Point #1: Supplementary Figure 4: To clearly validate the identification of glycosylation sites for readers, it would be beneficial to include MS/MS spectra with peak assignments for all glycosylation sites, especially for CbFla and CbFlaA5. Consider using formats similar to those presented in Figures 7C and 7D of the authors' previous study (Glycobiology, 31(3), 288-306, 2021).

Thank you for the suggestion. One representative MS/MS spectrum has been added for each glycosite in the constructs in **Supplementary figure 4 of the revised manuscript.**

Major Point #2: On page 3, the second paragraph of the introduction mentions that "...the Aeromonas caviae Maf glycosyltransferase modifies Ser/Thr residues flanked by hydrophobic amino acids such as leucine and isoleucine (28)." For CbMaf, is there a preference for any specific amino acid sequence around the glycosylated residues? If so, is there a correlation with physicochemical properties such as van der Waals volume, hydrophobicity, and electrical charge? Employing a tool like pLogo for statistical analysis of the glycosylation sites could be particularly beneficial (refer to Nature Methods, 10, 1211-1212, 2013).

Thank you for the suggestion. We have employed pLogo for analysis of the glycosylation sites in CbFla and we present the results in the revised manuscript, **lines 507-515, pg. 15**. Using the mass spectrometry data of the glycosites from this as well as previous studies (Twine et al, and Khairnar et al), we found a preponderance of small apolar or hydrophobic or polar uncharged residues (such as valine, methionine, isoleucine, phenylalanine, threonine, and glycine) immediately around the glycosite, flanked by polar charged or uncharged residues (such as arginine, lysine, aspartate, glutamate, asparagine, and glutamine) (**Supplementary data, Figures S4u-x**, in the revised manuscript).

Minor Points:

Minor Point #3: On page 3, the second paragraph of the introduction states, "...the A. caviae Maf glycosyltransferase interacts with both unglycosylated and glycosylated flagellin (29)." Please verify the accuracy of this citation.

Thank you for pointing this out. The citation has been corrected in the revised manuscript (**line 81, pg. 4**).

Minor Point #4: It would be helpful if Figures 1b, 2b, 2g, 2l, 3b, 3g, 3l, 4a, 4f, 4k, 4p, 5a, 5f, and 5k clearly indicated the identified glycosylation sites on the protein region diagrams.

Thank you for the suggestion. Since we do not have the data on the glycosites for all the constructs in the study, for the sake of uniform presentation, we have indicated the identified glycosylation sites, wherever available, in the domain architecture images provided in **Supplementary Figure S4** in the revised manuscript.

Minor Point #5: In Figures 1f, 2e, 2j, 2o, 3e, 3j, 3o, 4e, 4j, 4o, 4t, 5e, 5j, and 5o, substituting the labels "N" and "K" with "Neu5Ac" and "KDO," respectively, would clarify and prevent confusion with the potassium adduct.

Thank you for the suggestion. We have changed the letters N and K to Neu5Ac and KDO in all these figures (**Figures 1-5**) in the revised manuscript.

REVIEWERS' COMMENTS:

Reviewer #1 (Remarks to the Author):

My concerns have not been addressed, rather deferred as being beyond the scope of this study.

It can be argued that for engineering applications, glycosylation with the nonnative donor such as Neu5Ac and KDO could be interesting to warrant publication in a technologically-oriented journal. However, the contamination with KDO is indeed a problem when desiring Neu5Ac modified proteins. How will this be overcome? This promiscuity for Neu5Ac and KDO is certainly a consequence of the fact that the natural donor, most likely some derivative of legionaminic acid is not around, so the glycosyltransferase just grabs the next nonulosonic acid (Neu5Ac) octulosonic acid (KDO) that is around. This promiscuity for suboptimal donor glycans is also a big problem when Neu5Ac modifications are sought and therefore this system would only be useful in *E. coli* that produces Neu5Ac without KDO.

Overall, the engineering aspects such as the mini-flagellin constructs of this work could be useful, yet the KDO promiscuity limits the implementation value of this system at this stage.

From a biological perspective, without knowing the true donor glycan for the Cb Maf to enable comparisons with the efficiency of transfer and acceptor site specificity for Neu5Ac and KDO, the work unfortunately does not offer much new insight into the biology of Mafs. We still don't know how Mafs recognize the acceptor and this unknown is not developed and discussed in an insightful manner in the manuscript. Rather the interest of the authors seems to lie in the generation of mini-flagellin that still get modified in this specialised, non-native system and set-up and therefore I recommend publishing this work in a more technological or engineering journal.

Reviewer #3 (Remarks to the Author):

I thank the authors for seriously addressing my comments and those from other reviewers.

Point-by-point Responses to Referees' comments

We are very thankful to all three reviewers for the rigorous and thorough review of our manuscript and for the suggestions, which we feel have helped improve our manuscript. Our point-by-point responses to the reviewers' comments follow below.

Reviewer #1 (Expert in O-glycosylation of flagellin) (Remarks to the Author):

My concerns have not been addressed, rather deferred as being beyond the scope of this study.

It can be argued that for engineering applications, glycosylation with the nonnative donor such as Neu5Ac and KDO could be interesting to warrant publication in technologically-oriented journal. However the contamination with KDO is indeed a problem when desiring Neu5Ac modified proteins. How will this be overcome? This promiscuity for Neu5Ac and KDO is certainly a consequence of the fact that the natural donor, most likely some derivative of legionaminic acid is not around, so the glycosyltransferase just grabs the next nonulosonic acid (Neu5Ac) octulosonic acid (KDO) that is around. This promiscuity for suboptimal donor glycans is also a big problem when Neu5Ac modifications are sought and therefore this system would only be useful in E. coli that produces Neu5Ac without KDO.

Overall, the engineering aspects such as the mini-flagellin constructs of this work could be useful, yet the KDO promiscuity limits the implementation value of this system at this stage.

From a biological perspective, without knowing the true donor glycan for the Cb Maf to enable comparisons with the efficiency of transfer and acceptor site specificity for Neu5Ac and KDO, the work unfortunately does not offer much new insight into the biology of Mafs. We still don't know how Mafs recognize the acceptor and this unknown is not developed and discussed in an insightful manner in the manuscript. Rather the interest of the authors seems to lie in the generation of mini-flagellin that still get modified in this specialised, non-native system and set-up and therefore I recommend publishing this work in a more technological or engineering journal.

Thank you for raising the valid point that KDO modifications can be a problem when Neu5Ac modifications are sought. We have included this limitation of our experimental system in the discussion of the revised manuscript, and accordingly toned down our abstract and introduction.

Reviewer #3 (Remarks to the Author):

I thank the authors for seriously addressing my comments and those from other reviewers.

Thank you.